# BENCHMARKING PREDICTIVE CODING NETWORKS – MADE SIMPLE

**Luca Pinchetti**[1]**, Chang Qi**[2]**, Oleh Lokshyn**[2]**, Gaspard Oliviers**[3]**, Cornelius Emde**[1]**,**
**Mufeng Tang**[3]**, Amine M'Charrak**[1]**, Simon Frieder**[1]**, Bayar Menzat**[2]**,**
**Rafal Bogacz**[3]**, Thomas Lukasiewicz**[2,1]**, Tommaso Salvatori**[4,2*]

[1]Department of Computer Science, University of Oxford, Oxford, UK
[2]Institute of Logic and Computation, Vienna University of Technology, Vienna, Austria
[3]MRC Brain Network Dynamics Unit, University of Oxford, UK
[4]VERSES AI Research Lab, Los Angeles, US

## ABSTRACT

In this work, we tackle the problems of efficiency and scalability for predictive coding networks (PCNs) in machine learning. To do so, we propose a library, called PCX, that focuses on performance and simplicity, and use it to implement a large set of standard benchmarks for the community to use for their experiments. As most works in the field propose their own tasks and architectures, do not compare one against each other, and focus on small-scale tasks, a simple and fast open-source library and a comprehensive set of benchmarks would address all these concerns. Then, we perform extensive tests on such benchmarks using both existing algorithms for PCNs, as well as adaptations of other methods popular in the bio-plausible deep learning community. All this has allowed us to (i) test architectures much larger than commonly used in the literature, on more complex datasets; (ii) reach new state-of-the-art results in all of the tasks and datasets provided; (iii) clearly highlight what the current limitations of PCNs are, allowing us to state important future research directions. With the hope of galvanizing community efforts towards one of the main open problems in the field, scalability, we release code, tests, and benchmarks.[1]

## 1 INTRODUCTION

In 1999, Rao & Ballard (1999) proposed a formulation of predictive coding (PC) to model hierarchical information processing in the brain. It was recently realized that this framework could be used to train neural networks using a bio-plausible learning rule (Whittington & Bogacz, 2017). This has led to different research directions, whose focus was either to explore interesting properties of PC networks (Song et al., 2024; Alonso et al., 2022), or to propose variations that improve the performance on specific tasks (Salvatori et al., 2024; Ororbia & Kifer, 2022). These lines of research, however, have the tendency of not comparing their results against other works, and to focus on small-scale experiments. The field is hence avoiding what we believe to be the most important open problem: scalability.

There are multiple reasons why the problem of scalability has been overlooked. First, it is a hard problem, and it is still unclear why so far PC has been able to perform as well as classical gradient descent with backpropagation (BP) only up to a certain scale, which is that of small convolutional models trained to classify the CIFAR10 dataset (Salvatori et al., 2024). Understanding this would allow us to develop regularization techniques that stabilize learning, and hence allow better performance on more complex tasks. Second, the lack of specialized libraries makes PC models extremely slow: a full hyperparameter search on a small convolutional network can take several hours. Third, the lack of a common framework makes reproducibility and iterative contributions hard, as implementation details or code are rarely provided. In this work, we make the first steps toward addressing these problems with three contributions, that we call *tool*, *benchmarking*, and *analysis*.

---

*Corresponding author: `tommaso.salvatori@verses.ai`
[1]Link to the library: `https://github.com/liukidar/pcx`

**Tool.**  We release an open-source library for accelerated training for predictive coding called PCX. This library runs in JAX (Bradbury et al., 2018), and offers a user-friendly interface with a minimal learning curve through familiar syntax inspired by Pytorch. We also provide extensive tutorials. It is also fully compatible with Equinox (Kidger & Garcia, 2021), a popular deep-learning-oriented extension of JAX, ensuring reliability, extendability, and compatibility with ongoing research developments. It also supports JAX's Just-In-Time (JIT) compilation, making it efficient and allowing both easy development and execution of PC networks, gaining efficiency with respect to existing libraries.

**Benchmarking.**  We propose a uniform set of tasks, datasets, metrics, and architectures that should be used as a skeleton to test the performance of future variations of PC. The tasks that we propose are the standard ones in computer vision: image classification and generation. The models that we use, as well as the datasets, are picked according to two criteria: First, to allow researchers to test their algorithm from the easiest task (feedforward network on MNIST) to more complex ones; Second, to favor the comparison against related fields in the literature, such as equilibrium and target propagation (Scellier & Bengio, 2017; Bengio, 2014). To this end, we have picked some of the models that are consistently used in their research papers. As learning algorithms, we consider standard PC, incremental PC (Salvatori et al., 2024), PC with Langevin dynamics (Oliviers et al., 2024), and nudged PC, as done in the Eqprop literature (Scellier & Bengio, 2017; Scellier et al., 2024). Note that this is the first time nudging algorithms are applied in PC models.

**Analysis.**  We get state-of-the-art (SOTA) results for PC on multiple benchmarks and show for the first time that it is able to perform well on more complex datasets, such as CIFAR100 and Tiny Imagenet, where we get results comparable to those of backprop. In image generation tasks, we present experiments on datasets of colored images, going beyond MNIST and FashionMNIST as performed in previous works. We thoroughly discuss the results and highlight areas of improvement, the main one being generalization to very deep models, and report analysis on the credit assignment of PC in such cases, to better understand the reasons behind some failures. To conclude, in the supplementary material we provide a detailed explanation of hyperparameters/techniques/tricks that allowed us to reach SOTA results, to also provide a *cookbook* for researchers in the field.

## 2  RELATED WORKS

**Rao and Ballard's PC.**  The most related works are those that explore different properties or optimization algorithms of standard PC in the deep learning regime, using formulations inspired by Rao and Ballard's original work (Rao & Ballard, 1999). Examples are works that study their associative memory capabilities (Salvatori et al., 2021; Yoo & Wood, 2022; Tang et al., 2023; 2024), their ability to train Bayesian networks (Salvatori et al., 2022; 2023b), and theoretical results that explain, or improve, their optimization process (Millidge et al., 2022a;b; Alonso et al., 2022). Results in this field have allowed either to improve the performance of such models in different tasks, or to study different properties that could benefit from the use of PCNs.

**Variations of PC.**  In the literature, there are multiple variations of PC algorithms. Important examples are biased competition and divisive input modulation (Spratling, 2008), or the neural generative coding framework (Ororbia & Kifer, 2022). The latter is already used in multiple reinforcement learning and control tasks (Ororbia & Mali, 2023; Ororbia et al., 2023), and has its own JAX-based open source library called NGCLearn. For a review on how different PC algorithms evolved through time, from signal processing to neuroscience, we refer to (Spratling, 2017); for a more recent review specific to machine learning applications, to (Salvatori et al., 2023a). It is also worth mentioning the original literature on PC in the neurosciences has evolved from Rao and Ballard's work into a general theory that models information processing in the brain using probability and variational inference, called the *free energy principle* (Friston, 2005; Friston & Kiebel, 2009; Friston, 2010).

**Neuroscience-inspired deep learning.**  Another line of related works is that of neuroscience methods applied to machine learning, like equilibrium propagation (Scellier & Bengio, 2017), which is the most similar to PC (Laborieux & Zenke, 2022; Millidge et al., 2022a). Other methods able to train models of similar sizes are target propagation (Bengio, 2014; Ernoult et al., 2022; Millidge et al.,

2022b) and SoftHebb (Moraitis et al., 2022; Journé et al., 2022). The first two communities, that of targetprop and eqprop, consistently use similar architectures in their research papers to test their methods. In our benchmarking effort, some of the architectures proposed are the same ones, to favor a more direct comparison. There are also methods that differ more from PC, such as forward-only methods (Kohan et al., 2023; Nøkland, 2016; Hinton, 2022), and methods that back-propagate the errors using a designated set of weights (Lillicrap et al., 2014; Launay et al., 2020).

## 3 BACKGROUND AND NOTATION

Predictive coding networks (PCNs) are hierarchical Gaussian generative models that consist of $L$ levels. Each level models a multi-variate distribution, parameterized by the activation of the preceding level, which depends on both the model parameters $\theta = \theta_0, \theta_1, \theta_2, ..., \theta_L$ and the model state $h$. Let $h_l \in h$ be the realization of the vector of random variables $H_l$ of level $l$, then we have that the likelihood

$$P_\theta(h_0, h_1, \ldots, h_L) = P_{\theta_0}(h_0)P_{\theta_1}(h_1|h_0)\cdots P_{\theta_L}(h_L|h_{L-1}).$$

Where we write $P_{\theta_l}(h_l)$ instead of $P_{\theta_l}(H_l = h_l)$, that is the likelihood of $H_l$ evaluated at $h_l$. We refer to each of the scalar random variables of $H_l$ as a neuron. In PC both the prior on $h_0$ and the relationships between levels are governed by a normal distribution parameterized as follows:

$$P_{\theta_0}(h_0) = \mathcal{N}(h_0, \mu_0, \Sigma_0), \qquad \mu_0 = \theta_0,$$
$$P_{\theta_l}(h_l|h_{l-1}) = \mathcal{N}(h_l; \mu_l, \Sigma_l), \quad \mu_l = f_l(h_{l-1}, \theta_l),$$

where $\theta_l$ are the learnable weights parametrizing the transformation $f_l$, and $\Sigma_l$ is a covariance matrix, that will be fixed to the identity matrix throughout this work. If, for example, $\theta_l = (W_l, b_l)$ and $f_l(h_{l-1}, \theta_l) = \sigma_l(W_l h_{l-1} + b_l)$, then the neurons in level $l-1$ are connected to neurons in level $l$ via a linear operation, followed by a non-linear map, analogously to a fully connected layer. Intuitively, $\theta$ is the set of learnable weights of the model, while $h = \{h_0, h_1, ..., h_L\}$ is data-point-dependent latent state, containing the abstract representations for the given observations.

**Training.** In supervised settings, training consists of learning the relationship between given pairs of input-output observations $(x, y)$. In PC, this is performed by maximizing the joint likelihood of our generative model with the latent vectors $h_0$ and $h_L$ respectively fixed to the input and label of the provided data-point: $P_\theta(h|_{h_0=x,h_L=y}) = P_\theta(h_L = y, \ldots, h_1, h_0 = x)$. This is achieved by minimizing the so-called variational free energy $\mathcal{F}$ (Friston et al., 2007):

$$\mathcal{F}(h, \theta) = -\ln P_\theta(h) = -\ln\left(\mathcal{N}(h_0|\mu_0)\prod_{l=1}^{L}\mathcal{N}(h_l; f_l(h_{l-1}, \theta_l))\right) = \sum_{l=0}^{L}\frac{1}{2}(h_l - \mu_l)^2 + k. \quad (1)$$

The quantity $\epsilon_l = (h_l - \mu_l)$ is often referred to as *prediction error* of layer $l$, being the difference between the predicted activation $\mu_l$ and the current state $h_l$. For a full derivation of Eq. (1) we refer to the appendix. To minimize $\mathcal{F}$, the Expectation-Maximization (EM) (Dempster et al., 1977) algorithm is used by iteratively optimizing first the state $h$, and then the weights $\theta$ according to the equations

$$h^* = \mathrm{argmin}_h \mathcal{F}(h, \theta), \quad \theta^* = \mathrm{argmin}_\theta \mathcal{F}(h^*, \theta). \quad (2)$$

We refer to the first step described by Eq. (2) as *inference* and to the second as *learning* phase. In practice, we do not train on a single pair $(x, y)$ but on a dataset split into mini-batches that are subsequently used to train the model parameters. Furthermore, both inference and learning are approximated via gradient descent on the variational free energy. In the inference phase, firstly $h$ is *initialized* to an initial value $h^{(0)}$, and then, it is optimized for $T$ iterations. Then, during the learning phase, we use the newly computed values to perform a single update on the weights $\theta$. The gradients of the variational free energy with respect to both $h$ and $\theta$ are as follows:

$$\nabla h_l = \frac{\partial \mathcal{F}}{\partial h_l} = \frac{1}{2}\left(\frac{\partial \epsilon_l^2}{\partial h_l} + \frac{\partial \epsilon_{l+1}^2}{\partial h_l}\right), \qquad \nabla \theta_l = \frac{\partial \mathcal{F}}{\partial \theta_l} = \frac{1}{2}\frac{\partial \epsilon_l^2}{\partial \theta_l}. \quad (3)$$

Then, a new batch of data points is provided to the model and the process is repeated until convergence. As highlighted by Eq. (3), each state and each parameter is updated using local information as the gradients depend exclusively on the pre and post-synaptic errors $\epsilon_l$ and $\epsilon_{l+1}$. This is the main reason why, in contrast to BP, PC is a local algorithm and is considered more biologically plausible. In Appendix A, we provide an algorithmic description of the concepts illustrated in these paragraphs, highlighting how each equation is translated to code in PCX.

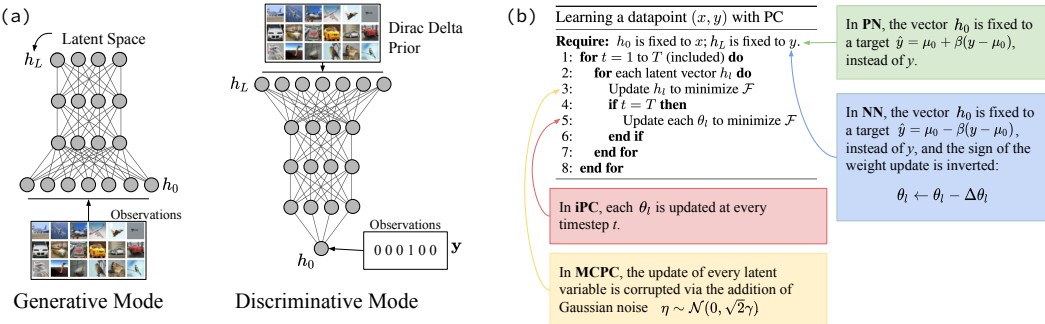

Figure 1: (a): Generative and discriminative modes; (b): Pseudocode of PC in supervised learning, where both the latent variables $h_l$ and the weight parameters $\theta_l$ are updated to minimize the variational free energy $\mathcal{F}$. In the colored boxes, informal description of the different algorithms considered in this work.

**Evaluation.** Given a test point $\bar{x}$, we fix $h_0 = \bar{x}$ and compute the most likely value of the latent states $h^*|_{h_0=\bar{x}}$, again using the state gradients of Eq. (3). We refer to this as *discriminative* mode. In practice, for discriminative networks, the values of the latent states computed this way are equivalent to those obtained via a *forward* pass, that is setting $h_l^{(0)} = \mu_l^{(0)}$ for every $l \neq 0$, as it corresponds to the global minimum of $\mathcal{F}$ (Frieder & Lukasiewicz, 2022).

**Generative Mode.** PCNs can also be used to perform unsupervised learning tasks. Given a data point $x$, the goal is to compress the information of $x$ into a latent representation, conceptually similar to how variational autoencoders work (Kingma & Welling, 2013). Such a compression is computed by fixing the state vector $h_L$ to the data point, and running inference – that is, we maximize $P_\theta(h|_{h_L=x})$ via gradient descent on $h$. The compressed representation will then be the value of $h_0$ at convergence (or, in practice, after $T$ steps). If we are training the model, we then perform a gradient update on the parameters to minimize the variational free energy of Eq. (1), as we do in supervised learning. A sketch of the discriminative and generative ways of training PCNs is represented in Fig. 1(a).

## 4 EXPERIMENTS AND BENCHMARKS

The benchmark that we propose is a standardized set of models, datasets, and testing procedures that have been consistently used to evaluate predictive coding, but in a non-uniform way. Here, for a comprehensive evaluation, we test models of increasing complexity on multiple computer vision datasets, with both feedforward and convolutional/de-convolutional layers; and multiple learning algorithms present in the literature. This section is divided into two areas that correspond to discriminative (supervised) and generative (unsupervised) inference tasks. For the former mode, we focus on supervised classification, and unsupervised generation for the latter. A sketch illustrating the two modes is in Fig. 1. For every class of experiments, we have performed a large hyperparameter search, and the details needed to reproduce the experiments, as well as a discussion about 'lessons learned' during such a large search, are in the Appendix B and C.

To provide a comprehensive evaluation, we have tested on multiple computer vision datasets, MNIST (LeCun & Cortes, 2010), FashionMNIST (Xiao et al., 2017), CIFAR10/100 (Krizhevsky et al., 2009), CelebA (Liu et al., 2018), and Tiny ImageNET (Le & Yang, 2015); on models of increasing complexity, and multiple learning algorithms present in the literature. The results, averaged over 5 seeds are reported in Tab. 1 when we used discriminative models, and in Tab. 2 for generative models. Note that, besides a very recent exception on CelebA (Sennesh et al., 2024), this is the first time that PCNs with local message passing are tested on datasets such as CelebA, CIFAR100, and Tiny ImageNet.

**Algorithms.** We consider various learning algorithms present in the literature: (1) Standard PC, already discussed in the background section; (2) Incremental PC (iPC) (Salvatori et al., 2024), a simple and recently proposed modification where the weight parameters are updated alongside the latent variables at every time step; (3) Monte Carlo PC (MCPC) (Oliviers et al., 2024), obtained by applying unadjusted Langevin dynamics to the inference process; (4) Positive nudging (PN), where the target used is obtained by a small perturbation of the output towards the original, 1-hot label; (5)

Table 1: Test accuracies of the different algorithms on different datasets.

| % Accuracy | PC-CE | PC-SE | PN | NN | CN | iPC | BP-CE | BP-SE |
|---|---|---|---|---|---|---|---|---|
| **MLP** | | | | | | | | |
| MNIST | $98.11^{\pm0.03}$ | $98.26^{\pm0.04}$ | $98.36^{\pm0.06}$ | $98.26^{\pm0.07}$ | $98.23^{\pm0.09}$ | $\mathbf{98.45^{\pm0.09}}$ | $98.07^{\pm0.06}$ | $98.29^{\pm0.08}$ |
| FashionMNIST | $89.16^{\pm0.08}$ | $89.58^{\pm0.13}$ | $89.57^{\pm0.08}$ | $89.46^{\pm0.08}$ | $89.56^{\pm0.05}$ | $\mathbf{89.90^{\pm0.06}}$ | $89.04^{\pm0.08}$ | $89.48^{\pm0.07}$ |
| **VGG-5** | | | | | | | | |
| CIFAR-10 | $86.61^{\pm0.14}$ | $87.98^{\pm0.11}$ | $88.42^{\pm0.66}$ | $88.83^{\pm0.04}$ | $\mathbf{89.47^{\pm0.13}}$ | $85.51^{\pm0.12}$ | $88.11^{\pm0.13}$ | $89.43^{\pm0.12}$ |
| CIFAR-100 (Top-1) | $60.00^{\pm0.19}$ | $54.08^{\pm1.66}$ | $64.70^{\pm0.25}$ | $65.46^{\pm0.05}$ | $\mathbf{67.19^{\pm0.24}}$ | $56.07^{\pm0.16}$ | $60.82^{\pm0.10}$ | $66.28^{\pm0.23}$ |
| CIFAR-100 (Top-5) | $84.97^{\pm0.19}$ | $78.70^{\pm1.00}$ | $84.74^{\pm0.38}$ | $85.15^{\pm0.16}$ | $\mathbf{86.60^{\pm0.18}}$ | $78.91^{\pm0.23}$ | $85.84^{\pm0.14}$ | $85.85^{\pm0.27}$ |
| Tiny ImageNet (Top-1) | $41.29^{\pm0.2}$ | $30.28^{\pm0.2}$ | $34.61^{\pm0.2}$ | $\mathbf{46.40^{\pm0.1}}$ | $46.38^{\pm0.11}$ | $29.94^{\pm0.47}$ | $43.72^{\pm0.1}$ | $44.90^{\pm0.2}$ |
| Tiny ImageNet (Top-5) | $66.68^{\pm0.09}$ | $57.31^{\pm0.21}$ | $59.91^{\pm0.24}$ | $68.50^{\pm0.18}$ | $69.06^{\pm0.10}$ | $54.73^{\pm0.52}$ | $\mathbf{69.23^{\pm0.23}}$ | $65.26^{\pm0.37}$ |
| **VGG-7** | | | | | | | | |
| CIFAR-10 | $84.62^{\pm0.1}$ | $81.91^{\pm0.3}$ | $85.97^{\pm0.3}$ | $87.26^{\pm0.1}$ | $88.40^{\pm0.12}$ | $80.15^{\pm0.18}$ | $88.60^{\pm0.1}$ | $\mathbf{89.91^{\pm0.1}}$ |
| CIFAR-100 (Top-1) | $56.80^{\pm0.14}$ | $37.52^{\pm2.60}$ | $56.56^{\pm0.13}$ | $59.97^{\pm0.41}$ | $64.76^{\pm0.17}$ | $43.99^{\pm0.30}$ | $59.96^{\pm0.10}$ | $\mathbf{65.36^{\pm0.15}}$ |
| CIFAR-100 (Top-5) | $83.00^{\pm0.09}$ | $66.73^{\pm2.37}$ | $81.52^{\pm0.17}$ | $81.50^{\pm0.41}$ | $84.65^{\pm0.18}$ | $73.23^{\pm0.30}$ | $\mathbf{85.61^{\pm0.10}}$ | $84.41^{\pm0.26}$ |
| Tiny ImageNet (Top-1) | $41.15^{\pm0.14}$ | $21.28^{\pm0.46}$ | $25.53^{\pm0.77}$ | $39.49^{\pm2.69}$ | $35.59^{\pm7.69}$ | $19.76^{\pm0.15}$ | $45.32^{\pm0.11}$ | $\mathbf{46.08^{\pm0.15}}$ |
| Tiny ImageNet (Top-5) | $66.25^{\pm0.11}$ | $44.92^{\pm0.27}$ | $50.06^{\pm0.84}$ | $64.66^{\pm1.95}$ | $59.63^{\pm6.00}$ | $40.36^{\pm0.22}$ | $\mathbf{69.64^{\pm0.18}}$ | $66.65^{\pm0.20}$ |
| **VGG-9** | | | | | | | | |
| CIFAR-10 | $78.12^{\pm0.14}$ | $75.33^{\pm0.25}$ | $76.90^{\pm0.18}$ | $85.90^{\pm0.14}$ | $87.19^{\pm0.41}$ | $79.02^{\pm0.21}$ | $89.18^{\pm0.08}$ | $\mathbf{90.02^{\pm0.18}}$ |
| CIFAR-100 (Top-1) | $58.25^{\pm0.13}$ | $39.57^{\pm0.18}$ | $43.21^{\pm0.21}$ | $60.74^{\pm0.75}$ | $58.92^{\pm1.61}$ | $44.76^{\pm0.40}$ | $60.63^{\pm0.28}$ | $\mathbf{65.51^{\pm0.23}}$ |
| CIFAR-100 (Top-5) | $83.28^{\pm0.06}$ | $66.90^{\pm0.26}$ | $71.13^{\pm0.23}$ | $83.19^{\pm0.38}$ | $81.56^{\pm0.63}$ | $72.88^{\pm0.29}$ | $\mathbf{85.25^{\pm0.11}}$ | $84.70^{\pm0.28}$ |
| Tiny ImageNet (Top-1) | $39.64^{\pm0.17}$ | $21.78^{\pm0.15}$ | $23.62^{\pm0.23}$ | $41.59^{\pm0.27}$ | $31.5^{\pm0.70}$ | $26.34^{\pm0.03}$ | $\mathbf{45.66^{\pm0.09}}$ | $45.51^{\pm0.15}$ |
| Tiny ImageNet (Top-5) | $64.60^{\pm0.09}$ | $44.43^{\pm0.09}$ | $46.89^{\pm0.11}$ | $66.15^{\pm0.32}$ | $54.67^{\pm0.68}$ | $50.48^{\pm0.05}$ | $\mathbf{69.65^{\pm0.09}}$ | $65.62^{\pm0.17}$ |
| ResNet-18 | | | | | | | | |
| CIFAR-10 | $43.19^{\pm0.61}$ | $53.74^{\pm0.43}$ | $62.45^{\pm0.52}$ | $62.33^{\pm0.93}$ | $55.29^{\pm1.65}$ | $70.44^{\pm0.81}$ | $92.83^{\pm0.18}$ | $\mathbf{93.21^{\pm0.07}}$ |
| CIFAR-100 (Top-1) | $16.01^{\pm0.42}$ | $22.83^{\pm0.38}$ | $25.86^{\pm0.86}$ | $26.91^{\pm0.55}$ | $15.45^{\pm1.7}$ | $29.45^{\pm1.36}$ | $\mathbf{72.32^{\pm0.26}}$ | $71.89^{\pm0.16}$ |
| CIFAR-100 (Top-5) | $40.67^{\pm0.70}$ | $50.18^{\pm0.52}$ | $53.80^{\pm1.13}$ | $55.57^{\pm0.80}$ | $39.42^{\pm2.8}$ | $56.70^{\pm1.73}$ | $\mathbf{92.14^{\pm0.12}}$ | $87.80^{\pm0.18}$ |
| Tiny ImageNet (Top-1) | $09.52^{\pm0.32}$ | $14.19^{\pm0.25}$ | $15.79^{\pm1.10}$ | $15.95^{\pm0.27}$ | $04.40^{\pm0.49}$ | $06.19^{\pm1.09}$ | $\mathbf{58.00^{\pm0.23}}$ | $55.30^{\pm0.16}$ |
| Tiny ImageNet (Top-5) | $26.21^{\pm0.50}$ | $34.55^{\pm0.20}$ | $37.36^{\pm1.57}$ | $37.76^{\pm0.52}$ | $14.30^{\pm1.92}$ | $16.51^{\pm3.09}$ | $\mathbf{79.94^{\pm0.06}}$ | $74.98^{\pm0.36}$ |

Negative nudging (NN), where the target is obtained by a small perturbation *away* from the target, and updating the weights in the opposite direction; (6) Centered nudging (CN), where we alternate epochs of positive and negative nudging (Scellier et al., 2024). Among these, PC, iPC, and MCPC will be used for the generative mode, and PC, iPC, PN, NN, and CN for the discriminative mode. See Fig. 1, and the supplementary material, for a more detailed description.

## 4.1 Discriminative Mode

We test the performance of PCNs on image classification tasks by comparing PC against BP, using both *Squared Error* (SE) and *Cross Entropy* (CE) loss, by adapting the energy function as described in Pinchetti et al. (2022). For the experiments on MNIST and FashionMNIST, we use feedforward models with 3 hidden layers of 128 hidden neurons, while for CIFAR10/100 and Tiny ImageNET, we compare ResNets and VGG-like models (He et al., 2016; Simonyan & Zisserman, 2014).

**Results.** Table 1 shows that the best performing algorithms, at least on the most complex tasks, are the nudging ones (PN, NN, and CN). Among them, CN is almost always the best performing one, a result that is in line with previous findings in the Eqprop literature (Scellier et al., 2024). The only case where nudging algorithms are outperformed is on Tiny Imagenet on VGG7, where PC-CE performs better than them. However, the results obtained by PC-CE are still worse than the ones obtained by CN on VGG5. The recently proposed iPC, on the other hand, performs well on small architectures, as it is the best performing one on MNIST and FashionMNIST, but its performance worsens when it comes to the training of large architectures. More broadly, the performance of models of depth up to 7 is comparable to those of backprop, while those of deeper models lag behind.

**Discussion on depth.** An interesting observation is that all the best results for PC have been achieved using a VGG5, with the performance trend being VGG5 > VGG7 > VGG9 > ResNet, as shown in Fig 2. Conversely, we observe the opposite for backprop-trained models, with deeper models like VGG9 outperforming VGG5. A similar trend was observed in ResNet18 experiments, where PCNs yielded significantly lower test accuracies, with none of the models coming close to the performance of a VGG5. In contrast, backprop-trained ResNet18 models outperformed all previously tested VGG models, further emphasizing the gap in scalability between the two. Future work should investigate the reason of such a phenomenon, as scaling up to more complex datasets will require the use of much deeper architectures. In Section 5, we analyze possible causes, as well as comparing the wall-clock time of the different algorithms.

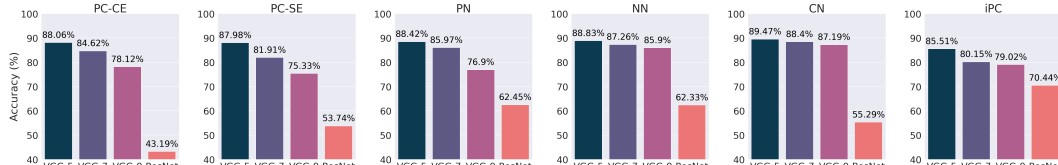

Figure 2: Test accuracies of different PC algorithms on the CIFAR10 dataset, using models of different depths.

Table 2: MSE loss for image reconstruction of BP, PC, and iPC on different datasets.

| MSE ($\times 10^{-3}$) | PC | iPC | BP | MSE ($\times 10^{-3}$) | PC | iPC | BP |
|---|---|---|---|---|---|---|---|
| MNIST | $9.25^{\pm 0.00}$ | $9.09^{\pm 0.00}$ | $\mathbf{9.08^{\pm 0.00}}$ | CIFAR-10 | $6.67^{\pm 0.10}$ | $\mathbf{5.50^{\pm 0.01}}$ | $6.17^{\pm 0.46}$ |
| FashionMNIST | $10.56^{\pm 0.01}$ | $10.11^{\pm 0.01}$ | $\mathbf{10.04^{\pm 0.00}}$ | CELEB-A | $2.35^{\pm 0.12}$ | $\mathbf{1.30^{\pm 0.12}}$ | $3.34^{\pm 0.30}$ |

## 4.2 GENERATIVE MODE

In this section, we test the performance of PCNs on image generation tasks. We perform three different kinds of experiments: (1) generation from a posterior distribution; (2) generation via sampling from the learned joint distribution; and (3) associative memory retrieval. In the first case, we provide a test image $y$ to a trained model, run inference to compute a compressed representation $\bar{x}$ (stored in the latent vector $h_0$ at convergence), and produce a reconstructed $\bar{y} = h_L$ by performing a forward pass with $h_0 = \bar{x}$. The models we consider have three layers, and we compare against autoencoders with a three-layer encoder/decoder structure (so, six layers in total). In the case of MNIST and FashionMNSIT we use feedforward layers, in the case of CIFAR10 and CelebA (de-)convolutional ones. The results in Tab. 2 and Fig. 3 report comparable performance, with a small advantage for PC compared to BP on the more complex tasks. In this case, iPC is the best performing algorithm, probably due to the small size of the considered models which allows for better stability.

Then, we tested the capability of PCNs to learn, and sample from, a complex probability distribution. MCPC extends PC by incorporating Gaussian noise to the activity updates of each neuron. This change enables a PCN to learn and generate samples analogous to a variational autoencoder (VAE). This change shifts the inference of PCNs from a variational approximation to Monte Carlo sampling of the posterior using Langevin dynamics. Data samples can be generated from the learned joint $P_\theta(h)$ by leaving all states $h_l$ free and performing noisy inference updates. Figure 4 illustrates MCPC's ability to learn multimodal distributions using the iris dataset (Pedregosa et al., 2011) and shows generative samples for MNIST. When comparing MCPC to a VAE, both models produced samples of similar quality. MCPC achieved a lower FID score (MCPC: $2.53^{\pm 0.17}$ vs. VAE: $4.19^{\pm 0.38}$), whereas the VAE attained a higher inception score (VAE: $7.91^{\pm 0.03}$ vs. MCPC: $7.13^{\pm 0.10}$).

In the associative memory (AM) experiments, we test how well the model is able to reconstruct a training image, after it is provided with an incomplete or corrupted version of it, as done in a previous work (Salvatori et al., 2021). Fig. 5 show the results obtained by a PCN with 2 hidden layers of 512 neurons given noise or mask corrupted images. In Tab. 3, we study the memory capacity as the number of hidden layers increases. No visual difference between the recall and original images can be observed for MSE up to 0.005. To evaluate efficiency we then trained

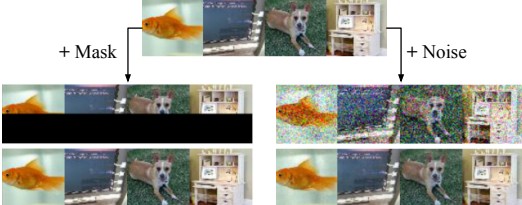

Figure 5: Memory recalled images. Top: Original images. Left: Noisy input (guassian noise, $\sigma = 0.2$) and reconstruction. Right: Masked input (bottom half removed) and reconstruction.

a PCN with 5 hidden layers of 512 neurons on 500 TinyImagenet samples, with a batch size of 50 and 50 inference iterations during training. Training takes $0.40 \pm 0.005$ seconds per epoch on an Nvidia V100 GPU.

**Discussion.** The results show that PC is able to perform generative tasks, as well as associative memory ones using decoder-only architectures. Via inference, PCNs are able to encode complex probability distributions in their latent state which can be used to perform a variety of different tasks, as we have shown. While this highlights the flexibility of PCNs when used in the generative mode, this comes at a higher computational cost due to the number of inference steps to perform.

Original Images

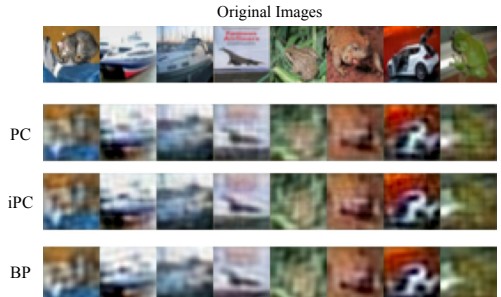

PC

iPC

BP

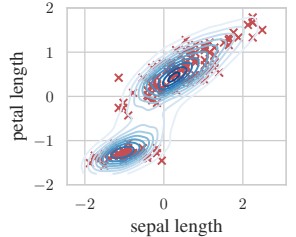

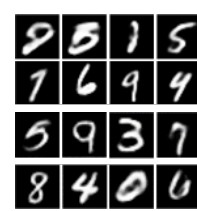

Figure 3: CIFAR10 image reconstruction via autoencoding convolutional networks. In order: original, PC, iPC, and BP.

Figure 4: Generative samples obtained by MCPC. Left: Contour plot of learned generative distribution compared to Iris data samples (x). Right: Samples obtained for a PCN. In order: unconditional generation, conditional generation (odd), conditional generation (even).

Table 3: MSE $(\times 10^{-4})$ of associative memory tasks given noisy (left) or masked (right) inputs as keys. Columns indicate the number of hidden neurons while rows shows the training images to memorize. Results over 5 seeds.

| Noise | 512 | 1024 | 2048 | | Mask | 512 | 1024 | 2048 |
|---|---|---|---|---|---|---|---|---|
| 50 | $6.06^{\pm 0.11}$ | $5.91^{\pm 0.14}$ | $5.95^{\pm 0.06}$ | | 50 | $0.06^{\pm 0.02}$ | $0.01^{\pm 0.00}$ | $0.00^{\pm 0.00}$ |
| 100 | $6.99^{\pm 0.19}$ | $6.76^{\pm 0.23}$ | $6.16^{\pm 0.07}$ | | 100 | $1.15^{\pm 0.78}$ | $1.01^{\pm 0.79}$ | $0.11^{\pm 0.03}$ |
| 250 | $9.95^{\pm 0.05}$ | $10.14^{\pm 0.06}$ | $8.90^{\pm 0.06}$ | | 250 | $39.1^{\pm 10.8}$ | $3.74^{\pm 0.73}$ | $0.22^{\pm 0.06}$ |

## 5 ANALYSIS AND METRICS

In this section, we report several metrics that we believe are important to understand the current state and challenges of training networks with PC and compare them with standard models trained with gradient descent and backprop when suitable. The first study we perform analyzes how the initialization of the network states $h$ influences the performance of the model. In the literature, they have been either initialized to be equal to *zero*, randomly initialized via a Gaussian prior (Whittington & Bogacz, 2017), or initialized via a forward pass. This last technique has been the preferred option in machine learning papers as it sets the errors $\epsilon_{l \neq L} = 0$ at every internal layer of the model. This allows the prediction error to be concentrated in the output layer only, and hence be equivalent to the SE. To provide a comparison among the three methods, we have trained a 3-layer feedforward model on FashionMNIST. The results, plotted in Fig. 6(a), show that forward initialization is indeed the better method, although the gap in performance shrinks the more iterations $T$ are performed.

**Energy propagation.** Concentrating the total error of the model to the last layer makes it hard for the inference process to then propagate such an energy back to the first layers. As reported in Fig. 6(b), we observe that the energy in the last layer is orders of magnitude larger than the one in the input layer, even after performing several inference steps. An easy way of quickly propagating the energy through the network would be to use learning rates equal to 1.0 for the updates of the states, that do not produce any energy imbalance, as also shown in Fig. 6(d). However, both the

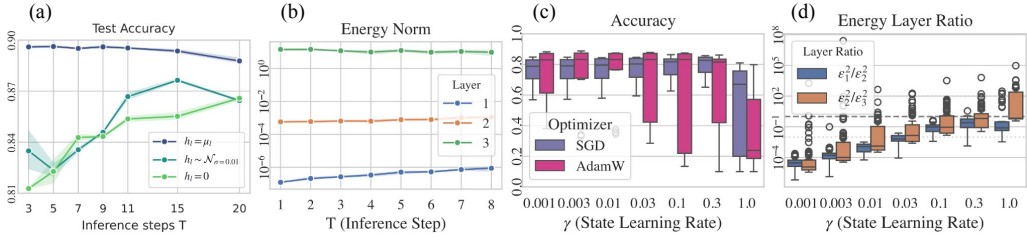

Figure 6: (a): Highest test accuracy reported for different initialization methods and iteration steps $T$ used during training; (b): Energies per layer during inference of the best performing model (which has $\gamma = 0.003$); (c) Decay in accuracy when increasing the learning rate of the states $\gamma$, tested using both SGD and Adam; (d) Imbalance between energies in the layers. Figures are obtained using a three layer model on FashionMNIST.

Table 4: Comparison of the training times of BP against PC on different architectures and datasets.

| Epoch time (seconds) | BP | PC (ours) | PC (Song) |
|---|---|---|---|
| MLP - FashionMNIST | $1.82^{\pm0.01}$ | $1.94^{\pm0.07}$ | $5.94^{\pm0.55}$ |
| AlexNet - CIFAR-10 | $1.04^{\pm0.08}$ | $3.86^{\pm0.06}$ | $17.93^{\pm0.37}$ |
| VGG-5 - CIFAR-100 | $1.61^{\pm0.04}$ | $5.33^{\pm0.02}$ | $13.49^{\pm0.05}$ |
| VGG-7 - Tiny ImageNet | $7.59^{\pm0.63}$ | $54.60^{\pm0.10}$ | $137.58^{\pm0.08}$ |

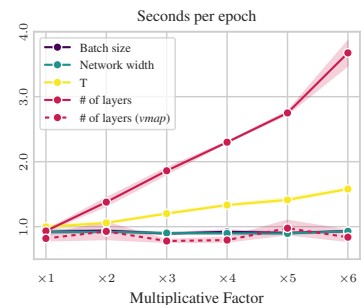

Figure 8: Training time for different network configurations.

results reported in Fig. 6(b), as well as our large experimental analysis of Section 4 show that the best performance was consistently achieved for state learning rates $\gamma$ significantly smaller than 1.0. This raises the question of whether better initialization or optimization techniques could result in a more balanced energy distribution and thus better weight updates.

To better understand how the energy propagation relates to the performance of the model, we have analyzed both the test accuracy and the ratio of the energies of subsequent layers as a function of the state learning rates $\gamma$. The results, reported in Fig 6(c,d), show that small learning rates lead to better performance, but also to large energy imbalances among layers. On the one hand, the energy in the first hidden layer is similar to that of the last layer for $\gamma = 1$, and about 6 orders of magnitude lower for $\gamma = 0.01$. On the other hand, models trained with a learning rate of $\gamma = 1$ achieve much worse performance. Such results show that the current training setup favors large energy imbalances among different layers, a problem that leads to exponentially small gradients when the depth of the model increases. We provide implementation details and results on other datasets in Appendix D.

**Training stability.** We have observed a link between the weight optimizer and the influence of the hidden dimension on the performance of the model. To better study this, we trained feedforward PCNs with different hidden dimensions, state learning rates $\gamma$ and optimizers, and reported the results in Fig. 7. The results show that, when using Adam, the width strongly affects the values of the learning rate $\gamma$ for which the training process is stable. Interestingly, this phenomenon does not appear when using both the SGD optimizer, nor on standard networks trained with backprop. This behavioral difference with BP is unexpected and suggests the need for better optimization strategies for PCNs, as AdamW was still the best choice in our experiments, but could

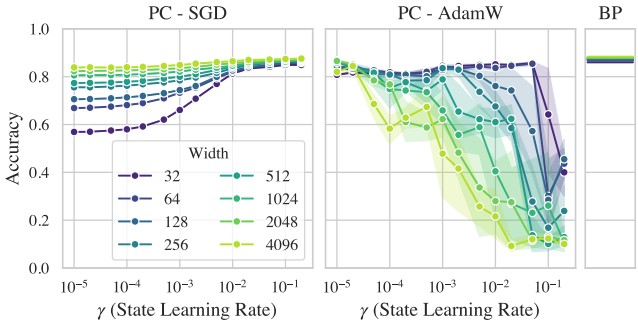

Figure 7: Updating weights with AdamW becomes unstable for wide layers as the accuracy plummets to random guessing for progressively smaller state learning rates as the network's width increases. Contrarily to using SGD, the optimal state learning rate depends on the width of the layers.

be a bottleneck for larger architectures.

## 6 LIBRARY, RESOURCES AND IMPLEMENTATIONS DETAILS

In this section, we discuss PCX, the tool that we have used to perform the experiments, and that we release open source. PCX is developed on top of JAX, focusing on performance and versatility, and is built upon the following concepts: *compatibility*, *modularity*, and *efficiency*.

**Compatibility.** PCX shares the same philosophy of equinox (Kidger & Garcia, 2021), according to which models are just PyTrees. Consequently, it is fully compatible, using a complete functional approach, with both libraries and many other tools developed for JAX, such as diffrax (Kidger, 2021) and optax (DeepMind et al., 2020). To this end, it will be straightforward to implement

novel development in deep learning into PCX. However, it also offers an imperative object-oriented interface, which allows researchers to build PCNs following a PyTorch-like style.

**Modularity.** Thanks to the object-oriented abstraction, we built the modular primitives that can be combined to create a PCN, mainly: a module class, representing abstract energy-based models; the vectorised nodes storing the states $h$; the optimizers, to perform the inference and learning process in a predictive coding network; and various standard *Layers*. Each benchmark we showcase in this work can be obtained by combining and configuring different *blocks* as needed.

**Efficiency.** PCX extensively relies on just-in-time compilation. From our initial benchmarks, we observed a speed-up of up to 50x when compiling a PCN. We believe that this stark difference is due to the nature of PC, which relies on multiple smaller operations compared to backpropagation, i.e., the $T$ inference step performed in each layer, and thus is more affected by the function calls overhead present in eager execution mode.

PCX offers a unified interface to test multiple variations of PC on several tasks. Our modular code base can easily be expanded in the future to support new variations of PC, as we show complete compatibility with existing variations and training techniques. This is different from, for example, the monolithic or low-level approaches used in (Song, 2024) and (Ororbia & Kifer, 2022), respectively.

### 6.1 COMPUTATIONAL RESOURCES AND LIMITATIONS.

We measured the wall-clock time of our PCNs implementation against another existing open-source library (Song, 2024) used in many PC works (Song et al., 2024; Salvatori et al., 2021; 2022; Tang et al., 2023), as well as comparing it with equivalent BP-trained networks (developed also with PCX for a fair comparison). Tab. 4 reports the measured time per epoch, averaged over 5 trials, using a A100 GPU. We also outperform alternative methods such as Eqprop: using the same architecture on CIFAR100, the authors report that one epoch takes $\approx 110$ seconds, while we take $\approx 5.5$ on the same hardware (Scellier et al., 2024). However, this is not an apple-to-apple comparison, as the authors are more concerned with simulations on analog circuits, rather than achieving optimal GPU usage.

**Limitations.** The efficiency of PCX could be further increased by fully parallelizing all the operations. In fact, in its current state, JIT is unable to parallelize the execution of the layers; a problem that can be addressed with the JAX primitive *vmap*, but only in the unpractical case where all the layers have the same dimension. To test how different hyperparameters of the model influence the training speed, we have taken a feedforward model, and trained it multiple times, each time increasing a specific hyperparameter by a multiplicative factor. The results, reported in Fig. 8, show that the two parameters that increase the training time are the number of layers $L$ and the number of steps $T$. Ideally, only $T$ should affect the training time as inference is an inherently sequential process that cannot be parallelized, but this is not the case, as the time scales linearly with the amount of layers. Details are reported in Appendix G.

## 7 DISCUSSION

The main contribution of this work is the introduction and open-source release of PCX, a library that can be used to perform deep learning tasks using PCNs. Its efficiency relies on JAX's Just-In-Time compilation and carefully structured primitives built to take advantage of it. A second advantage of our library is its intuitive setup, tailored to users already familiar with other deep learning frameworks such as PyTorch. This, together with the large number of tutorials we release, will make it easy for new users to train networks using PC. We have then used PCX to perform an extensive comparative study among different models and training algorithms present in the literature, obtained by testing a large number of parameter combinations and activation functions.

In terms of results, we have shown that predictive coding networks perform comparably to standard deep learning ones trained with BP, conditioned on the fact that small/medium size architectures are used, such as VGG 7. When this condition is relaxed, the performance of predictive coding fails to match that of BP, able to scale along with model size. In the supplementary material, we add rigorous studies that provide more details about how the energy flows inside PCNs over time, and their training stability, as well as show how PCNs classify out-of-distribution data, and possible solutions for training extremely deep networks via the use of skip connections.

ACKNOWLEDGMENTS

We thank the reviewers for their valuable feedback and insightful discussions, which have significantly enhanced this manuscript. Amine M'Charrak gratefully acknowledges support from the Evangelisches Studienwerk e.V. Villigst through a doctoral fellowship. Tommaso Salvatori gratefully acknowledges funding from VERSES AI. Rafal Bogacz gratefully acknowledged support by Medical Research Council grant MC_UU_00003/1. Cornelius Emde is supported by the EPSRC Centre for Doctoral Training in Health Data Science (EP/S02428X/1). Thomas Lukasiewicz and Cornelius Emde are supported by the AXA Research Fund.

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

# Contents

## SOCIETAL IMPACT

This work adheres to the established ethical standards prevalent in the field of AI and machine learning. In the short term, it does not introduce specific ethical concerns, as the models and technology we study are still in early-stage development, and do not perform as well as classic methods. However, we acknowledge the implications and responsibilities that accompany advancements in these technologies. We are committed to ongoing evaluation and responsible stewardship of our contributions to ensure they align with the ethical landscape of this dynamic field.

APPENDIX

Here we provide the details on how experiments were conducted and results obtained. We opt for a more descriptive approach to convey the fundamental concepts, and leave all details for reproducibility in the provided code, as well as in the next sections. There, each section will link to the exact directory corresponding to the described experiments.

## A  PCX – A BRIEF INTRODUCTION

In this section, we illustrate the core ideas of PCX by describing the main building blocks necessary to train and evaluate a feedforward classifier in predictive coding. For more detailed and complete explanations, please refer to the tutorial notebooks in the *examples* folder of the library.

In Section 3, we defined PCNs as models with parameters $\theta = \{\theta_0, \ldots, \theta_L\}$ and state $h = \{h_0, \ldots, h_L\}$. In PCX, we divide a model in two main components: *layers* (i.e., the traditional deep-learning transformations such as 'Linear' or 'Conv2D') and *vodes* (i.e., vectorized nodes that store the array of neurons representing state $h_l$). A PCN is defined as follows:

```python
import jax.nn as jnn
import pcx.predictive_coding as pxc
import pcx.nn as pxnn

class MLP(pcx.EnergyModule):
    def __init__(self, in_dim, h_dim, out_dim):
        self.layers = [
            pxnn.Linear(in_dim, h_dim),
            pxnn.Linear(h_dim, h_dim),
            pxnn.Linear(h_dim, out_dim)
        ]

        self.vodes = [
            pxc.Vode((dim,)) for dim in (h_dim, h_dim, out_dim)
        ]

    def __call__(self, x, y = None):
        for layer, vode in zip(self.layers, self.vodes):
            u = jnn.leaky_relu(layer(x))
            x = vode(u)

        if y is not None:
            self.vodes[-1].set("h", y)

        return u
```

In the *__call__* method, we forward the input $x$ through the network. Note that every time we call a vode, we are effectively storing in it the activation $u_l$ (so that we can later compute the energy $\epsilon_l^2$ associated to the vode) and return its state $h_l$ (i.e., *x = vode(u)* corresponds to *vode.set("u", u); x = vode.get("h")*). During training, the label $y$ is provided to the model and fixed to the last vode by overwriting its state $h^{(L)}$. Note that, since both during training and evaluation the state of the first vode would be fixed to the input $x$, we avoid defining it (i.e., we avoid computing $P_{\theta_0}(h_0)$ since it would be constant), and directly forward $x$ to the first layer transformation.

The class *pxc.EnergyModule* provides a *.energy()* function that computes the variational free energy $\mathcal{F}$ as per Eq. (1). We can compute the state and parameters gradients as per Eqs. (3) by calling *pxf.value_and_grad*, a wrap around the homonymous JAX function. Having defined two optimizers, *optim_w* and *optim_h*, for parameters and state respectively, we can define training on a pair $(x, y)$ as following:

```python
import pcx.utils as pxu
import pcx.functional as pxf
```

```python
def energy(x, y, *, model):
    model(x, y)
    return model.energy()

grad_h = pxf.value_and_grad(
    pxu.Mask(pxc.VodeParam, [False, True])
)(energy)

grad_w = pxf.value_and_grad(
    pxu.Mask(pxc.LayerParam, [False, True])
)(energy)

def train(T, x, y, *, model, optim_h, optim_w):
    model.train()

    # Initialization
    with pxu.step(model, pxc.STATUS.INIT, clear_params=pxc.VodeParam.Cache):
        model(x)

    # Inference steps
    for i in range(T):
        with pxu.step(model, clear_params=pxc.VodeParam.Cache):
            _, g_h = grad_h(x, y, model=model)
            optim_h.step(model, g_h["model"], True)

    # Learning step
    with pxu.step(model, clear_params=pxc.VodeParam.Cache):
        _, g_w = grad_w(x, y, model=model)
        optim_w.step(model, g_w["model"])
```

A few notes on the above code:

- JAX (Bradbury et al., 2018) is a functional library, PCX is not. Modules in PCX are PyTrees, using the same philosophy as another popular JAX library, equinox (Kidger & Garcia, 2021), with which PCX modules are fully compatible. However, their state is managed by PCX so that each parameter transformation is automatically tracked. The user can opt in for this behavior by passing arguments as keyword arguments (such as in the above example). Positional function parameters, instead are ignored by PCX and it is the user's duty to track their state as done in JAX or equinox.

- *pxf.value_and_grad* allows to specify a *Mask* object to identify which parameters to target with the given transformation. In the case above, we first compute the gradient of $\mathcal{F}$ with respect of the state (*VodeParam*) and, then, of the weights (*LayerParam*) of the model.

- In the *train* function, we use *pxu.step* to set the model status to *pxc.STATUS.INIT* to perform the state initialization. In PCX, forward initialization is the default method, however other ones can be easily specified. *pxu.step* is also used to clear the PCN's cache which is used to store intermediate values such as the activations $u_l$.

- The actual examples in the library are on mini-batches of data, so all transformations above are *vmapped* in the actual experiments.

For the evaluation function, being in discriminative mode, we simply perform a forward pass through the PCN which sets $\epsilon_l = 0$ for all layers.

```python
def eval(x, *, model):
    with pxu.step(model, pxc.STATUS.INIT, clear_params=pxc.VodeParam.Cache):
        return model(x)
```

## B    DISCRIMINATIVE EXPERIMENTS

**Model.**    We conducted experiments on three models: MLP, VGG-5, and VGG-7. The detailed architectures of these models are presented in Table 5.

Table 5: Detailed Architectures of base models

|  | **MLP** | **VGG-5** | **VGG-7** |
|---|---|---|---|
| Channel Sizes | [128, 128] | [128, 256, 512, 512] | [128, 128, 256, 256, 512, 512] |
| Kernel Sizes | - | [3, 3, 3, 3] | [3, 3, 3, 3, 3, 3] |
| Strides | - | [1, 1, 1, 1] | [1, 1, 1, 1, 1, 1] |
| Paddings | - | [1, 1, 1, 0] | [1, 1, 1, 0, 1, 0] |
| Pool window | - | $2 \times 2$ | $2 \times 2$ |
| Pool stride | - | 2 | 2 |

For each model, we conducted experiments with the following different algorithms:

1. Standard PC with Cross-Entropy Loss (**PC-CE**) / Mean Squared Error Loss (**PC-SE**): already discussed in the background section.

2. PC with Positive Nudging (**PC-PN**):

   Unlike standard Predictive Coding with Mean Squared Error Loss (PC-SE), where the output is clamped to the target, we "nudge" the output towards the target in PC with nudging. This is achieved by fixing the representation $h$ of last layer $h_L$ to $\mu_L + \beta(y - \mu_L)$, where $\mu_L$ is the predicted activation of the last layer after forward initialisation, $y$ is the target, and $\beta \in (0, 1)$ is a scalar parameter that controls the strength of nudging. Note that when $\beta = 1$, PC with nudging is equivalent to the standard PC.

   During training procedure, as the model output gradually approaches to the target, we employ a strategy of increasing $\beta$. At the end of each epoch, the value of $\beta$ is incremented by a fixed rate $\beta_{ir}$. When $\beta$ becomes greater than or equal to 1, we set it to 1. This strategy allows the model more stable to learn and explore in the early stages of training, while gradually transitioning to the standard PC in the later stages.

3. PC with Negative Nudging (**PC-NN**):

   In this algorithm, we do the opposite of positive nudging: we push the output away from the target. Therefore, we fix the representation $h$ of the last layer to $\mu_L - \beta(y - \mu_L)$. We use the same strategy of dynamically increasing $\beta$. When $\beta$ becomes greater than or equal to -1, we set it to 1.

   In the learning stage, to ensure that the direction of the weight update is consistent with the target (since we fixed $h_L$ to the opposite direction), we invert the weight update: $\theta_l \leftarrow \theta_l - \Delta\theta_l$ where $\Delta\theta_l$ defined in the Eq. (3).

4. PC with Center Nudging (**PC-CN**):

   Center Nudging (Scellier et al., 2024) is used in equilibrium propagation to improve and stabilize performance compared to both positive and negative nudging, and it is obtained as an average of the gradients produced by the two methods. Here, we approximate this behavior by randomly alternating between epochs in which we train with either negative or positive nudging. In this way, the training model can benefit from both methods without any extra computational cost.

5. Incremental PC (**iPC**), a simple and recently proposed modification where the weight parameters are updated alongside the latent variables at every time step (Salvatori et al., 2024).

6. Standard Backpropagation with Cross-Entropy Loss (**BP-CE**) / Mean Squared Error Loss (**BP-SE**): the most popular way to do the credit assignment in the neural networks. The model is trained by computing the gradients of the loss function with the weights of the network using the chain rule.

**Experiments.** The benchmark results of MLP are obtained with MNIST and Fashion-MNIST, the results of VGG-5 are obtained with CIFAR-10, CIFAR-100 and Tiny ImageNet, the results of VGG-7 are obtained with CIFAR-100 and Tiny ImageNet. The data is normalized as in Table 6.

Table 6: Data normalization

|  | **Mean ($\mu$)** | **Std ($\sigma$)** |
|---|---|---|
| MNIST | 0.5 | 0.5 |
| Fashion-MNIST | 0.5 | 0.5 |
| CIFAR-10 | [0.4914, 0.4822, 0.4465] | [0.2023, 0.1994, 0.2010] |
| CIFAR-100 | [0.5071, 0.4867, 0.4408] | [0.2675, 0.2565, 0.2761] |
| Tiny ImageNet | [0.485, 0.456, 0.406] | [0.229, 0.224, 0.225] |

For data augmentation on the training sets of CIFAR-10, CIFAR-100, and Tiny ImageNet, we apply random horizontal flipping with a probability of 50%. Additionally, we employ random cropping with different settings for each dataset. For CIFAR-10 and CIFAR-100, images are randomly cropped to 32×32 resolution with a padding of 4 pixels on each side. In the case of Tiny ImageNet, random cropping is performed to obtain 56×56 resolution images without any padding. And on the testing set of Tiny ImageNet, we use center cropping to extract 56×56 resolution images, also without padding, since the original resolution of Tiny ImageNet is 64x64.

The model hyperparameters are determined using the search space shown in Table 7. The results presented in Table 1 were obtained using 5 seeds with the optimal hyperparameters.

As for the optimizer and scheduler, we use mini-batch gradient descent (SGD) with momentum as the optimizer for the $h$, and we utilize AdamW Loshchilov & Hutter (2017) with weight decay as the optimizer for the $\theta$. Additionally, we apply a warmup-cosine-annealing scheduler without restart for the learning rates of $\theta$.

Table 7: Hyperparameters search configuration

| **Parameter** | **PC** | **iPC** | **BP** |
|---|---|---|---|
| Epoch (MLP) | | 25 | |
| Epoch (VGG and ResNet) | | 50 | |
| Batch Size | | 128 | |
| Activation | [leaky relu, gelu, hard tanh] | | [leaky relu, gelu, hard tanh, relu] |
| $\beta$ | [0.0, 1.0], 0.05[1] | - | - |
| $\beta_{ir}$ | [0.02, 0.0] | - | - |
| $lr_h$ | (1e-2, 5e-1)[2] | (1e-2, 1.0)[2] | - |
| $lr_\theta$ | (1e-5, 3e-4)[2] | | (3e-5, 3e-4)[2] |
| $momentum_h$ | [0.0, 1.0], 0.05[1] | | - |
| $weightdecay_\theta$ | (1e-5, 1e-2)[2] | (1e-5, 1e-1)[2] | (1e-5, 1e-2)[2] |
| T (MLP and VGG-5) | [4,5,6,7,8] | | - |
| T (VGG-7) | [8,9,10,11,12] | | - |
| T (VGG-9) | [9,10,12,15,18] | | - |
| T (ResNet-18) | [6,10,12,18,24] | | - |

[1]: "[a, b], c" denotes a sequence of values from a to b with a step size of c.
[2]: "(a, b)" represents a log-uniform distribution between a and b.

**Results.** All the results presented in this study were obtained using forward initialization, a technique that initializes the model's parameters by performing a forward pass on a zero tensor with the same shape as the input data. Besides, in our experiments, we limited the range of $T$ to ensure a fair comparison with BP in terms of training times. Higher $T$ correspond to a greater number of optimization rounds of $h$, which can lead to improved model performance but also increased computational costs and longer training durations. To maintain comparability with BP, we restricted our searching space of T that resulted in training times similar to those observed in BP-based training.

**Momentum helps significantly.** In Figure 9, we present the accuracy of the VGG-7 model trained on CIFAR-100 using different momentum values, both without nudging(Figure 9a) and with nudging(Figure 9b). It is evident from Figure 9 that selecting an appropriate momentum value can substantially improve model accuracy. By comparing Figures 9a and 9b, we can observe that different training algorithms have different optimal momentum values. The optimal momentum for training with nudging is generally higher than that for training without nudging. Furthermore, the optimal momentum for negative nudging is larger than that for positive nudging. These differences in optimal momentum values highlight the importance of carefully tuning the momentum hyperparameter based on the specific training algorithm and nudging method employed. For reference, the optimal model parameters and momentum values for various tasks and models can be found in the example/discriminative_experiments folder of the PCX library.

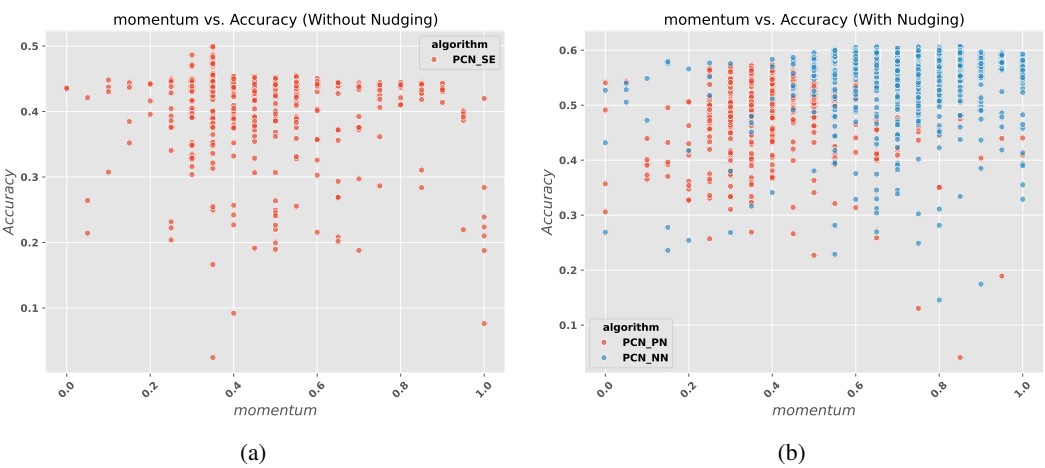

(a)                                                                 (b)

Figure 9: Comparison of the accuracy of the VGG-7 model trained on CIFAR-100 using different momentum values

**Activation function also plays a crucial role in improving model accuracy.** For models using Cross-Entropy Loss, the "HardTanh" activation function is a better choice. In the case of models using Mean Squared Error Loss without nudging, the "LeakyReLU" activation function tends to perform better. When using Positive Nudging, the optimal activation function varies depending on the model architecture. For Negative Nudging, the "GeLU" activation function is the most suitable choice.

**Nudging improves performance.** Fig. 10 illustrates the relationship between the learning rate of $h$ and accuracy with or without nudging. From the plot, we can observe that when nudging is not used (red dots), the model achieves better results at lower learning rates. However, when nudging is employed (purple and blue dots), regardless of whether it is positive nudging or negative nudging, the model can attain better accuracy at higher learning rates compared to the case without nudging. Additionally, Fig. 9b shows the relationship between momentum and accuracy. We can see that after applying nudging, the model can achieve better results at higher momentum values. We believe this is the reason why nudging can improve performance. The ability to use higher learning rates and momentum values without sacrificing accuracy is a significant advantage of nudging, as it can lead to faster convergence and improved generalization performance.

## C  GENERATIVE EXPERIMENTS

### C.1  AUTOENCODER

An Autoencoder is a network that learns how to compress a high-dimensional input into a much smaller dimensional space, called the bottleneck dimension or the hidden dimension, as accurately as possible. Thus, a backpropagation-based Autoencoder consists of two parts: an encoder, that compresses the input from the original high-dimensional space into the bottleneck dimension, and a decoder, that reconstructs the original input from the bottleneck dimension. A mean-squared error

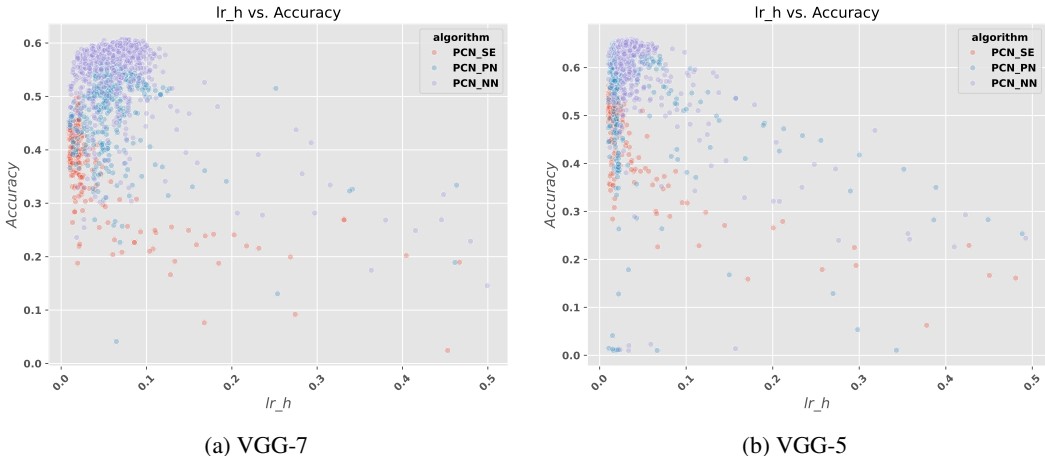

(a) VGG-7            (b) VGG-5

Figure 10: Comparison of the accuracy of the VGG-7 and VGG-5 model trained on CIFAR-100 using different learning rates for $h$.

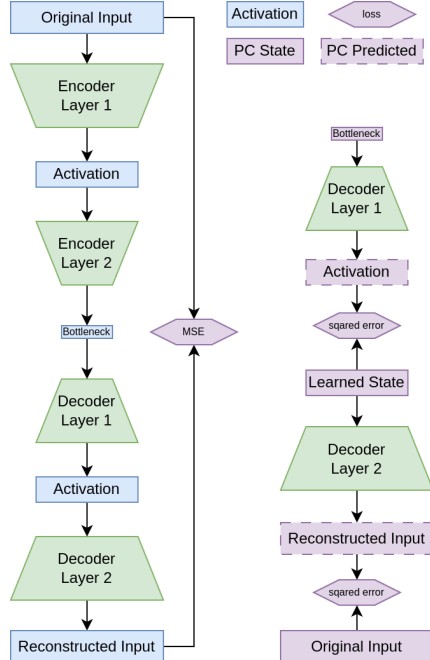

Figure 11: **Left.** An Autoencoder implemented with backpropagation consists of both an encoder and a decoder. The encoder compresses the input data into the bottleneck dimension, and the decoder restores the original image. **Right.** An Autoencoder implemented with Predictive Coding. The state of the first PC layer is the bottleneck dimension. The state of the last PC layer is the original input, and the predicted state of the last PC layer is the predicted input. Inference steps update the bottleneck dimension to make it a good compressed representation.

(MSE) between the original and the reconstructed input is used as a loss to train the Autoencoder network in an unsupervised manner.

Predictive Coding (PC) alleviates the need in the encoder part of an Autoencoder. Specifically, only the decoder part of an Autoencoder is used, with a PC layer acting as the bottleneck dimension and as an input to the decoder. Moreover, PC layers are inserted after each layer of the decoder.

A PC-based Autoencoder works as follows:

Table 8: Hyperparameters and search spaces for deconvolution-based autoencoders

| Parameter | PC | iPC | BP |
|---|---|---|---|
| Number of layers | 3 | | conv layers: 3 deconv layers: 3 |
| Internal state dimension | | 4x4 | |
| Internal state channels | | 8 | |
| Kernel size | | [3, 4, 5, 7] | |
| Activation function | | [relu, leaky_relu, gelu, tanh, hard_tanh] | |
| Batch size | | 200 | |
| Epochs | | 30 | |
| T | 20 | | - |
| Optim $h$ | SGD+momentum | | - |
| $lr_h$ | $(1e\text{-}2, 5e\text{-}1)^2$ | $(1e\text{-}2, 1.0)^2$ | - |
| $momentum_h$ | [0.0, 0.95] | | - |
| Optim $\theta$ | | AdamW | |
| $lr_\theta$ | | $3e\text{-}5, 1e\text{-}3^2$ | |
| $weightdecay_\theta$ | $(1e\text{-}5, 1e\text{-}2)^2$ | $(1e\text{-}5, 1e\text{-}1)^2$ | $(1e\text{-}5, 1e\text{-}2)^2$ |

Table 9: Hyperparameters and search spaces for linear-based autoencoders

| Parameter | PC | iPC | BP |
|---|---|---|---|
| Number of layers | 3 | | encoder: 3 decoder: 3 |
| Internal state dimension | | 64 | |
| Activation function | | [relu, leaky_relu, gelu, tanh, hard_tanh] | |
| Batch size | | 200 | |
| Epochs | | 30 | |
| T | 20 | | - |
| Optim $h$ | SGD+momentum | | - |
| $lr_h$ | $(1e\text{-}2, 5e\text{-}1)^2$ | $(1e\text{-}2, 1.0)^2$ | - |
| $momentum_h$ | [0.0, 0.95] | | - |
| Optim $\theta$ | | AdamW | |
| $lr_\theta$ | | $(3e\text{-}5, 1e\text{-}3)^2$ | |
| $weightdecay_\theta$ | $(1e\text{-}5, 1e\text{-}2)^2$ | $(1e\text{-}5, 1e\text{-}1)^2$ | $(1e\text{-}5, 1e\text{-}2)^2$ |

1. The energy function of the last PC layer is set to MSE upon its creation. In PCX, the squared error is the default energy function. The squared error is then summed across all dimensions in the input and averaged over the batch, that approximates the MSE up to a multiplication constant.

2. The current state of the last PC Layer $L$, $h_L$, is fixed to the original input data, which means that $h_L$ is not changed during inference steps.

3. Since the energy of the last layer $L$ now encodes the MSE loss between the predicted image $\mu_L$ and the original input stored as $h_L$, the inference steps will update the current states $h_l$ of all PC layers but the last one, including the one that represents the bottleneck dimension, to minimize this MSE loss.

4. Once the inference steps are done, the state of the bottleneck dimension PC layer will converge to the compressed representation of the original input.

## C.2 MCPC

**Model.** Monte Carlo predictive coding (MCPC) is a version of predictive coding that can be used for generative learning. MCPC differs from PC by its noisy neural dynamics. Unlike PC where the

neural activity converges to a mode of the free-energy, the neural activity of MCPC performs noisy gradient descent which is used for Monte Carlo sampling. When an input is provided, the noisy neural activity samples the posterior distribution of the generative model given the sensory input. When no input is provided the neural activity samples the generative model encoded in the model parameters. Specifically, the neural dynamics of MCPC leverage the following Langevin dynamics:

$$\Delta h_l = -\gamma \nabla_{h_l} \mathcal{F}_{h_l}(h, \theta) + \sqrt{2\gamma} N \tag{4}$$

where N is a Gaussian random variable with variance $\sigma^2_{mcpc}$. These neural dynamics can be extended to 2nd-order Langevin dynamics for faster sampling:

$$\Delta h_l = \gamma r_l \tag{5}$$

$$\Delta r_l = \gamma \nabla_{h_l} \mathcal{F}(h, \theta) - \gamma(1 - m)r_l + \sqrt{2(1 - m)\gamma} N \tag{6}$$

where $m$ is a momentum constant.

An MCPC model is trained following a Monte Carlo expectation maximisation scheme which iterates over the following two steps: (i) MCPC's neural activity samples the model's posterior distribution for the given data, and (ii) the model parameters are updated to increase the model log-likelihood under the samples of the posterior. In practice, we run MCPC inference for a limited number of steps after which we update the model parameters with a single sample of the posterior similarly to how model parameters are updated in variational auto encoders.

After training, samples of a trained model are generated by leaving all neurons unclamped and recording the activity of input neurons (the neurons clamped to data during training). The activity is recorded after a limited number of activity update steps. This process is repeated for each data sample.

MCPC's implementation in PCX utilizes a noisy SGD optimizer for the state $h$. Compared to PC than uses an SGD or Adam optimizer, MCPC incorporates an optimizer that merges the addition of noise to the model's gradients with an SGD optimizer. The variance of the noise added to the gradients needs to be carefully crafted to scale appropriately with the learning rate and the momentum as shown in equations (4 - 6).

**Experiments.** All the MCPC experiments use feedforward models with Squared Error (SE) loss. The SE loss of the state layer $h_L$ is also scaled by a variance parameter $\sigma^2_{h_L}$. This additional parameter is introduced to prevent the Gaussian layer $h_L$ from having a variance much larger than the variance of the data which would prevent learning. Moreover, for unconditional learning and generation, the layer $h_0$ is left unclamped during both training and generation. In contrast, for the conditional learning task on MNIST, the layer $h_0$ is clamped to labels during training and generation.

For the iris dataset, we train a model with layer dimensions [2 x 64 x 2], tanh activation function and default parameter values (state learning rate $\gamma$=0.01, state `momentum` = 0.9 , noise state variance $\sigma^2_{mcpc} = 1$, parameter learning rate $lr_\theta$, parameter decay = 0.0001, Adam parameter optimizer, layer variance $\sigma^2_{h_L} = 0.01$ and a batch size of 150). We use 500 state update steps during learning and 10000 for generation.

For the unconditional learning task on MNIST, we train models with layer dimensions [30 x 256 x 256 x 256 x 784]. The model hyperparameters for MCPC and VAE were determined using the hyperparameter search shown in table 10 to optimize the FID and the inception score separately. Refer to the code for exact optimal parameter values. We use 1000 state update steps during learning and 10000 for generation.

For the conditional learning task on MNIST, we train models with layer dimensions [2 x 256 x 256 x 256 x 784]. The labels used in this task, clamped to $h_0$, specify whether an image corresponds to an even or odd number. The model hyperparameters are determined using the search space shown in table 10. We use 1000 state update steps during learning and 10000 for generation.

**Results.** Figure 12 shows samples generated by the trained models for hyperparameters that maximize the inception score.

## C.3 ASSOCIATIVE MEMORIES

This section describes the experimental setup of associative memory tasks.

Table 10: Bayes hyperparameter search configuration for MCPC and VAE (where applicable) on MNIST.

| Parameter | Value |
|---|---|
| activation | {ReLU, Silu, Tanh, Leaky-ReLU, Hard-Tanh} |
| $\gamma$ | log-uniform(0.0001, 0.05) |
| momentum | {0.0, 0.9} |
| $\sigma^2_{mcpc}$ | {1.0, 0.3, 0.01, 0.001} |
| $lr_\theta$ | log-uniform(0.0001, 0.1) |
| parameter decay | {0.0, 0.1, 0.01, 0.001, 0.0001} |
| $\sigma^2_{h_L}$ | log-uniform(0.03, 1.0) |
| batch size | {150, 300, 600, 900} |

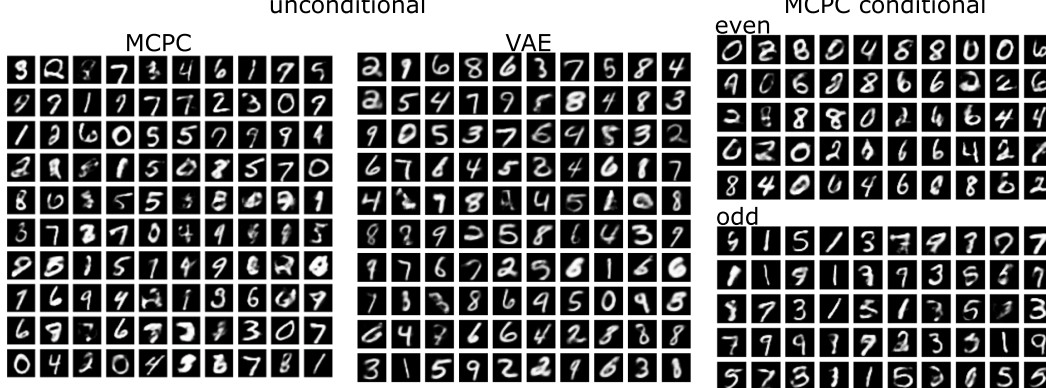

Figure 12: Samples generated by trained models that optimize the inception score under the unconditional and conditional learning regimes.

**Model.** A generative PCN is first trained on $n$ images sampled from the Tiny ImageNet dataset until its parameters have converged. Then, a corrupted version of the training images is presented to the sensory layer of the model ($h_L$) and we run inference $\nabla h_l$ on all layers, including the sensory layer, until convergence. Note that in masked experiments, the intact top half of the images is kept fixed during inference. Intuitively, suppose the model has minimized its free energy with its sensory layer fixed at each of the $n$ training examples during training. In that case, it has formed attractors defined by these training examples and would thus tend to "refine" the corrupted images to fall back into the energy attractors.

**Experiments.** Here, the benchmark results are obtained with Tiny ImageNet, corrupted with either Gaussian noise with 0.2 standard deviation, or a mask on the bottom half of the images (examples shown in Fig. 5). We vary the model size and number of training examples to memorize, to study the capacity of the models. Specifically, we use a generative PCN with architecture $[512, d, d, 12288]$ where $d = [512, 1024, 2048]$ (12288 being the flattened Tiny ImageNet images) and varied $n = [50, 100, 250]$. We performed a hyperparameter search for each $d$ and $n$ on the parameter learning rate $lr_\theta \in \{1 \times 10^{-4} + k \cdot 5 \times 10^{-5} \mid k \in \mathbb{Z}, 0 \leq n \leq 18\}$, the state learning rate $\gamma \in \{0.1 + k \cdot 0.05 \mid k \in \mathbb{Z}, 0 \leq n \leq 18\}$, training inference steps $T_{\text{train}} \in [20, 50, 100]$ and recall inference steps $T_{\text{recall}} \in [50000, 100000]$. We fix the activation function of the model to Tanh, and the number of training epochs to $500$ and a batch size of $50$. The results in Table 3 are obtained with $5$ seeds with the searched optimal hyperparameters.

## D ENERGY AND STABILITY

This section describes the experimental setup of Section 5, provides replications on other datasets and ablations.

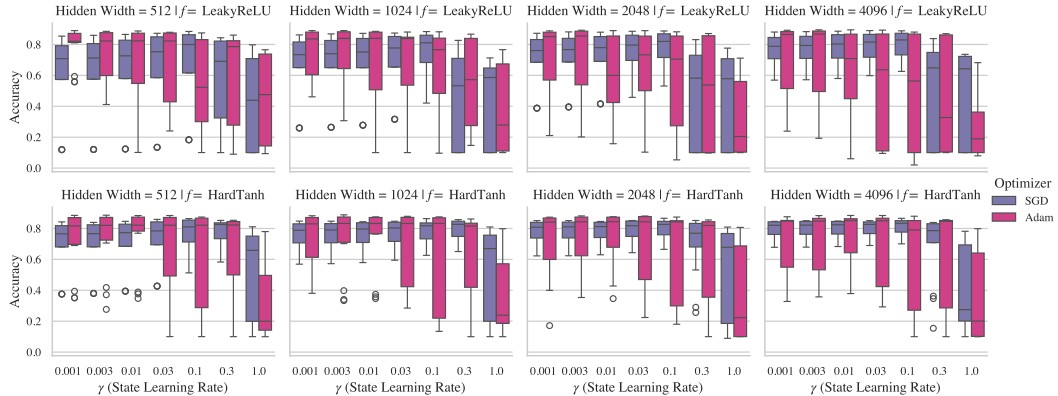

Figure 13: Model accuracies for a range of combinations of activation functions and model widths. Adam perfers small learning rates and tends to be less stable than SGD. Obtained on FashionMNIST.

## D.1  ENERGY PROPAGATION

We test a grid of models on multiple datasets to examine the energy propagation in the models. We test on the FashionMNIST, Two Moons, and, Two Circles datasets. The Two Circles dataset is particularly interesting, as poor energy distribution intuitively results in a linear inductive bias (we primarily learn a one-layer network). This linear inductive bias harms the performance on Two Circles (linear model accuracy $\approx 50\%$) more than FashionMNIST ($\approx 83\%$) and Two Moons ($\approx 86\%$).

**Experimental Setup.**   We train a grid of feedforward PCNs with 2 hidden layers. We train on three datasets: FahionMNIST (as reported in the main body) and additionally Two Moons and Two Circles. For all models, we train for 8 epochs with $T = 8$ inference steps. States are optimized with SGD and forward initialization. The grid is formed over weight learning rate $lr_\theta \in \{1 \times 10^{-5}, 1 \times 10^{-4}, \dots, 1\}$, state learning rate $\gamma \in \{1 \times 10^{-3}, 3 \times 10^{-3}, 1 \times 10^{-2}, 3 \times 10^{-2}, 1 \times 10^{-1}, 3 \times 10^{-1}, 1\}$, activation functions $f \in \{\text{LeakyReLU}, \text{HardTanh}\}$ (the former is unbounded the latter is bounded), optimization with AdamW or SGD with momentum $m \in \{0.0, 0.5, 0.9, 0.95\}$ and hidden widths of $\{512, 1024, 2048, 4096\}$ for FashionMNIST and $\{128, 256, 512, 1024\}$ for Two Moons and Two Cricles.  We replicate all experiments on 3 seeds for FashionMNIST and 10 seeds for the other datasets.

**Results.**   Fig. 6(left) in the main paper shows the average energy across the last batch at the end of training for the best performing model on the grid. Fig. 6(center-left) compares SGD with momentum 0.9 and AdamW. It is obtained for activation function "HardTanh" and a width of 1024. We replicate this figure for the other combinations of activation functions and widths below in Fig 13. We observe that across all conditions, small to medium state learning rates are generally preferred by SGD, while AdamW has a stronger preference to smaller state learning rates. Given the uneven distribution of energies across layers, AdamW, in particular, may not scale to deeper architectures. We further, observe a larger variance in performance for AdamW, especially for wider layers, which we discuss in paragraph "Training Instability" in Sec. 5 and below. Fig. 6(right) is based on all models trained with AdamW. Many models with high state learning rates diverge, we only plot models achieving accuracy $> 0.5$.

Below we present the results of experiments on the Two Moons and Two Circles datasets. Fig. 14b, 14a, and 14c replicateFig. 6 for Two Moons, and Fig. 15b, 15a, and 15c for Two Circles. Results are very similar to FashionMNIST: The energy is concentrated in the last layer, even after $T$ inference steps. However, in the example for Two Circles, we actually observe a training effect for earlier layers: While the energy increases first due to error propagation (still orders of magnitude below later layers), the energy is reduced afterwards. Energy ratios are consistently indicating poor energy propagation for state learning rates $\gamma$, that perform well. As predicted the variance in results is significantly larger for Two Circles, especially for small state learning rates.

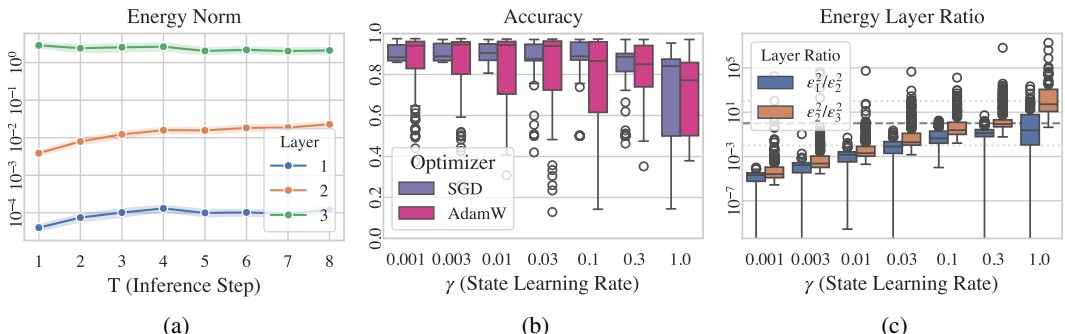

Figure 14: Energy propagation on the Two Moons dataset. 14a shows the imbalance between layers across $T$ steps. 14b shows the model performance across state learning rates and 14c the energy distribution across state learning rates.

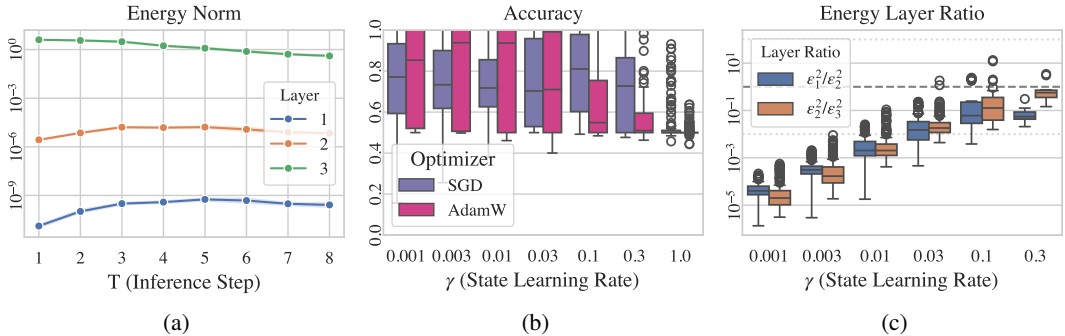

Figure 15: Energy propagation on the Two Circles dataset. 15a shows the imbalance between layers across $T$ steps. 15b shows the model performance across state learning rates and 15c the energy distribution across state learning rates.

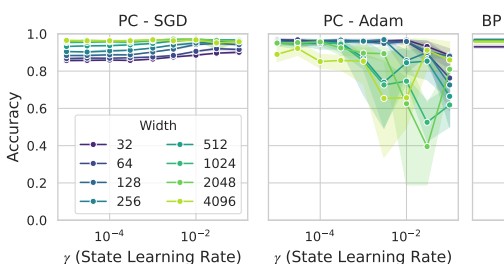
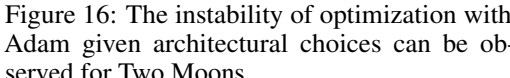
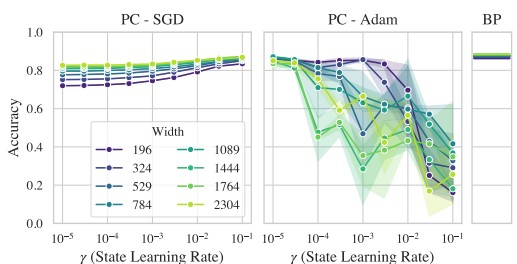

Figure 16: The instability of optimization with Adam given architectural choices can be observed for Two Moons.

Figure 17: The instability of optimization as a result of an optimizer-architecture-interaction can be (at least partially) be attributed to the *absolute* size of layers.

## D.2 TRAINING STABILITY

We test a grid of PCNs to analyze the interaction between model width, state learning rates and weight optimizers.

**Experimental Setup.** We train models on FashionMNIST (as reported above) and Two Moons. We train feedforward PCNs (2 hidden layers) with "LeakyReLU" activations over a grid of parameters. All models are trained over 8 epochs. The widths of the hidden layers are $\{32, 64, \ldots, 4096\}$. State variables are trained for $T = 8$ steps with SGD and learning rates $\gamma \in \{1 \times 10^{-5}, 3 \times 10^{-5}, \ldots, 0.3\}$. The weights are updated through SGD or the Adam optimizer with a learning rate of 0.01 for FashionMNIST and 0.03 for Two Moons. Both optimizers uses 0.9 momentum for weights. We further train baseline BP models with the same hyperparameters. For FashionMNIST we replicate each run over 3 random initializations, for Two Moons over 10.

**Results.** We replicate Fig. 7 (FashionMNIST) here for the Two Moons dataset, see Fig. 16. We observe effects for Two Moons that are analog to FashionMNIST as presented above: The stability of optimization strongly depends on the width of the hidden layers for Adam. This effect is not observed for SGD on either dataset. This further supports the our conclusion in Sec. 5: While Adam is the better optimizer, this interaction effect (width $\times$ $\gamma$) can hinder the scaling of PCNs with Adam. Optimization methods for PCNs require further attention from the research community.

**Ablation.** We further provide an ablation on FashionMNIST. In the experiments above, the hidden layer width is altered, introducing changes in the *absolute size* of the hidden layers (i.e. number of neurons), but also changing the *relative size* of the hidden layers in the network, as input and output layers remain the same size across all experiments. Hence, we provide another experiment on FashionMNIST, where we increase the image size and augment the label vector with 0s, such that the width of all layers is equal. All other experimental variables remain as described above. The results are shown in Fig. 17 and follow the trend observed in Fig. 7 and 16: We find that there exists an interaction between the optimization and the width of the network as described above. Hence, accounting for relative changes in layer width does *not* sufficiently explain the problem and we conclude that the *absolute* size of the layers plays a role in the stability of optimization with AdamW.

**ResNets** Here we discuss the findings on the energy propagation in light of the ResNets18 experiments. In this section, we have shown that lower learning rate for the nodes harm energy propagation, and that the AdamW optimizer displays poor performance for larger hidden dimensions. To this end, we have trained ResNets18 using SGD and large learning rates for the nodes, and compared the performance against those in the main body of the paper. The performance are, however, not comparable to the ones reported in Table.1, as ResNets trained with SGD on the CIFAR10 dataset reach accuracies of 39.9% and 43.2% when using PC and iPC, respectively. To better understand the incidence of different hyperparameters on the final test accuracy of the models, in Fig. 18 we show their importance plots. Such quantities are computed by fitting a random forest regressor with hyperparameters as datapoints, accuracies as labels, and extracting the feature importance.

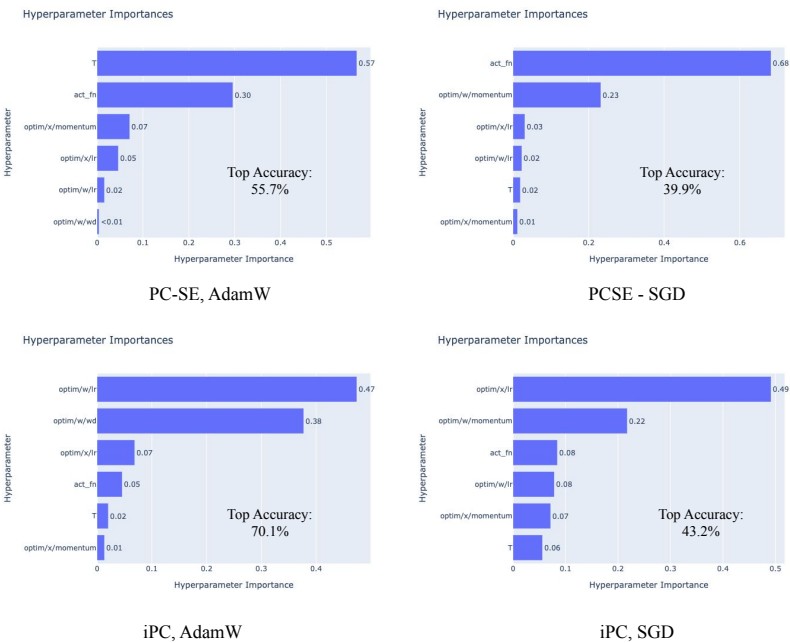

Figure 18: Importance plots that show the importance of each hyperparameter in the final test accuracy of the model, computed by fitting a random forest regressor with hyperparameters as datapoints, accuracies as labels, and extracting the feature importance.

# E    SKIP CONNECTIONS INTO VGG19

**Skip connections.**    We investigate the integration of skip connections into the VGG19 architecture to enhance its performance on the CIFAR10 image classification task, showing a significant increase in test accuracy from 25.32% to 73.95%. The vanishing gradient problem, a notable challenge in deep Predictive Coding (PC) models, becomes pronounced with increased network depth, hindering error transmission to earlier layers and impacting learning efficacy. To address this, we introduce skip connections that allow gradients to bypass multiple layers, enhancing gradient flow and overall learning performance.

Table 11: Hyperparameter configuration and best accuracy for VGG19 with and without skip connections on CIFAR10

| Parameter | Range | Best Value |
|---|---|---|
| **With Skip Connections** | | |
| Epochs | 30 | 30 |
| Batch size | 128 | 128 |
| Activation functions | {GELU, Leaky ReLU} | Leaky ReLU |
| Optimizer for network parameters - Learning rate | {5e-2, 1e-1, 5e-1} | 0.5 |
| Optimizer for network parameters - Momentum | {0.0, 0.5, 0.9, 0.99} | 0.5 |
| Optimizer for weight parameters - Learning rate | 1e-4 | 1e-4 |
| Optimizer for weight parameters - Weight decay | {5e-4, 1e-4, 5e-5} | 5e-4 |
| Number of inference steps (T) | {24, 36} | 24 |
| **Best Accuracy** | | **73.95%** |
| **Without Skip Connections** | | |
| Epochs | 30 | 30 |
| Batch size | 128 | 128 |
| Activation functions | {GELU, Leaky ReLU} | GELU (default) |
| Optimizer for network parameters - Learning rate | {5e-2, 1e-1, 5e-1} | 0.1 |
| Optimizer for network parameters - Momentum | {0.0, 0.5, 0.9, 0.99} | 0.99 |
| Optimizer for weight parameters - Learning rate | 1e-4 | 1e-4 |
| Optimizer for weight parameters - Weight decay | {5e-4, 1e-4, 5e-5} | 1e-4 |
| Number of inference steps (T) | {24, 36} | 24 |
| **Best Accuracy** | | **25.32%** |

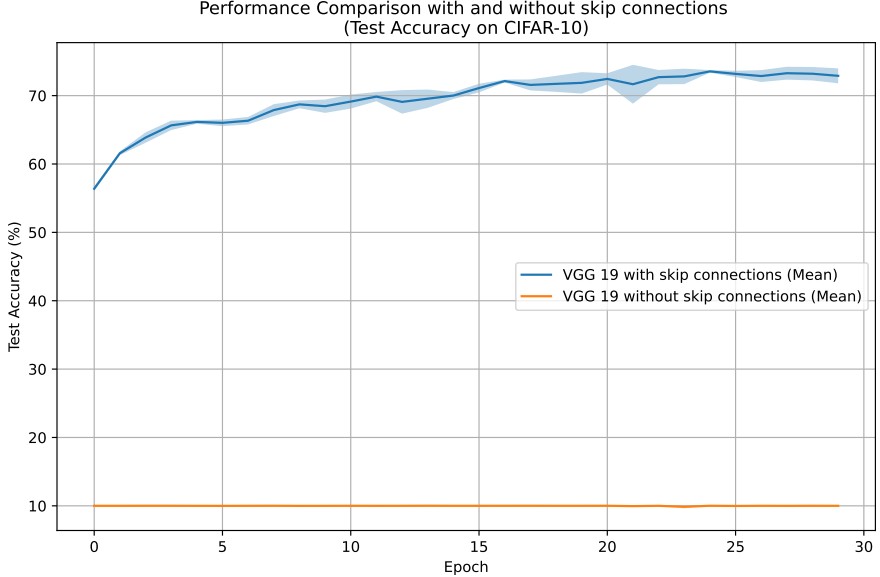

Figure 19: Performance comparison of VGG19 with and without skip connections on the CIFAR-10 dataset over 30 epochs. The plot shows the mean test accuracy along with the shaded area representing the variability across three different seeds.

**Results** Our modified VGG19 model includes a skip connection from an early layer within the feature extraction stage, with the output flattened and adjusted using a linear layer before being

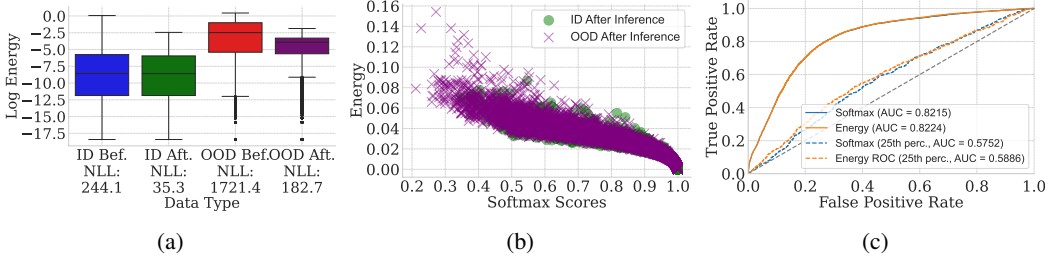

Figure 20: (a) Energy and NLL of ID/OOD data before and after state optimization. (b) Nonlinearity between energy and softmax post-convergence. (c) ROC curve of OOD detection at the 100th and 25th percentiles of scores. In all plots, "ID" refers to MNIST and "OOD" to FashionMNIST.

reintegrated during the classification stage. The model underwent rigorous training and evaluation on the CIFAR10 dataset, employing standard preprocessing techniques like normalization and data augmentation (horizontal flips and rotations). Detailed hyperparameter tuning revealed optimal configurations for both models, with and without skip connections, exploring various optimizers, learning rates, momentum values, and weight decay settings, significantly enhancing the model performance with skip connections as summarized in Table 11. Figure 19 shows the test accuracy progression over 30 epochs for the VGG19 model with and without skip connections on the CIFAR10 dataset, using three different seed values and identical hyperparameters for both simulations.

## F    PROPERTIES OF PREDICTIVE CODING NETWORKS

This section describes the experimental setup of Section F.1 and displays the utility of using the free energy of a PCN classifier to differentiate between in-distribution (ID) and out-of-distribution (OOD) data (Liu et al., 2020). We show how one can compute the negative log-likelihood of various datasets (Grathwohl et al., 2020) under the PCN. We further provide analyses on the relationship between maximum softmax values and energy values before convergence and after convergence at the state optimum. We compare results across multiple datasets to corroborate our results as well as to show how PCNs can be used for OOD detection out of the box based on a single trained PCN classifier for which we study the receiver operating characteristic (ROC) curve based on different percentiles of the softmax and energy scores.

### F.1    FREE ENERGY AND OUT-OF-DISTRIBUTION DATA.

With PCX, it is straightforward to inspect and analyze several properties of PCNs. Here, we use $\mathcal{F}$ to differentiate between in-distribution (ID) and out-of-distribution (OOD) due to a semantic distribution shift (Liu et al., 2020), as well as to compute the likelihood of a datasets (Grathwohl et al., 2020). This can occur when samples are drawn from different, unseen classes, such as FashionMNIST samples under an MNIST setup (Hendrycks & Gimpel, 2017).

**Experimental Setup.**    We train a PCN classifier on MNIST using a feedforward PCNs with 3 hidden layers each of size $H = 512$ with "GELU" activation and cross entropy loss in the output layer. We train the model until test error convergence using early stopping at epoch 75. During training the state variables are optimized for $T = 10$ steps with SGD and state learning rate $\gamma = 0.01$ without momentum. The weights are optimized using the SGD optimizer with a momentum of $m_\theta = 0.9$ and the weight learning rate is chosen as $lr_\theta = 0.01$. During test-time inference, we optimize the state variables until convergence for $T = 100$. To understand the confidence of a PCN's predictions, we compare the distribution of energy for ID and OOD samples against the distribution of the softmax scores that the classifier generates. We compute negative log-likelihoods for ID and OOD samples under the PCN classifier via:

$$\mathcal{F} = -\ln p(x, y; \theta) \implies p(x, y; \theta) = e^{-\mathcal{F}}, \tag{7}$$

We conduct the experiments on MNIST as the in-distribution (ID) dataset and we compare it against various out-of-distribution datasets such as notMNIST, KMNIST, EMNIST (letters) as well as FashionMNIST.

Briefly, the results in Fig. 20a demonstrate that a trained PCN classifier can effectively (1) assess OOD samples out-of-the-box, without requiring specific training for that purpose (Yang et al., 2021), and (2) produce energy scores for ID and OOD samples that initially correlate with softmax values prior to the optimization of the states variables, $h$. However, after optimizing the states for $T$ inference steps, the scores for ID and OOD samples become decorrelated, especially for samples with lower softmax values as shown in Fig. 20b. To corroborate this observation, we also present ROC curves for the most challenging samples, including only the lowest $25\%$ of the scores. As shown in Fig.20c, the probability (i.e., energy-based) scores provide a more reliable assessment of whether samples are OOD. Experiment details and results on other datasets are provided in in Appendix F. Additional, and more detailed results for the EMNIST (letters) and KMNIST datasets are provided below.

**Results.**    In the following we briefly interpret the additional results on the basis of experiments supported by various figures

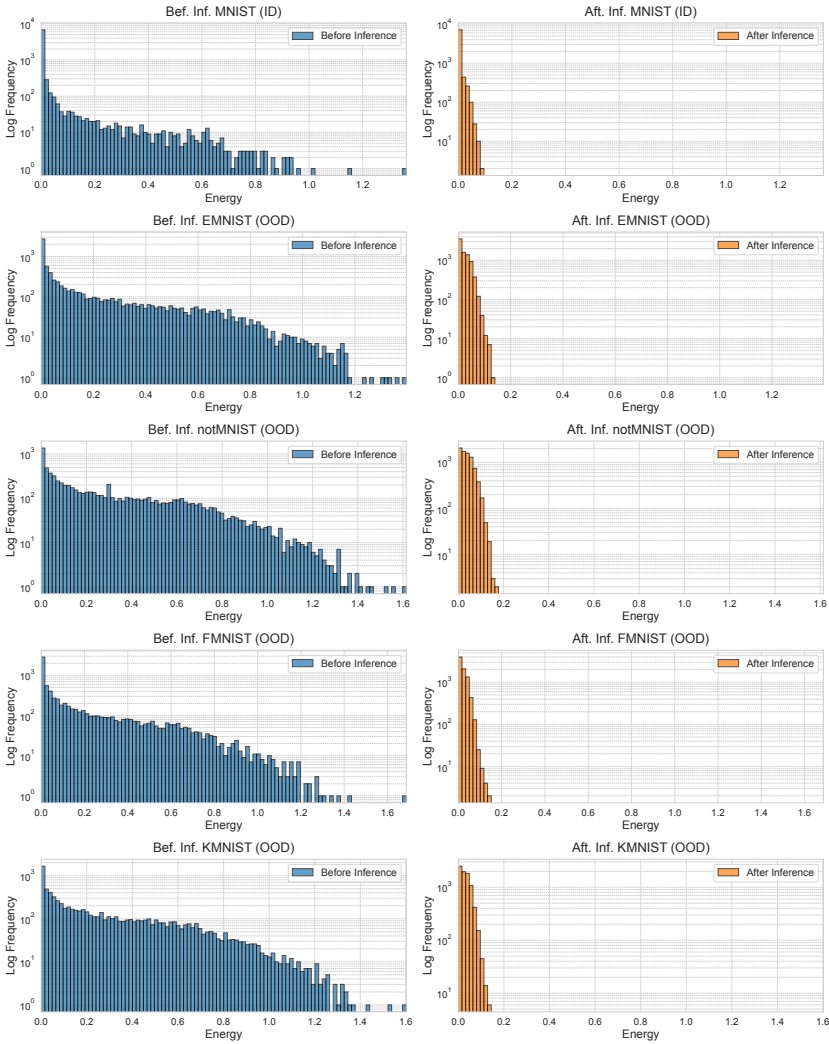

Figure 21: Energy distributions before and after state optimization.

In Fig. 21 we see how the energy is distributed at test-time before and after state optimization. We can see, that all OOD datasets have significantly larger initial energies as well as final energies compared to the ID dataset (MNIST).

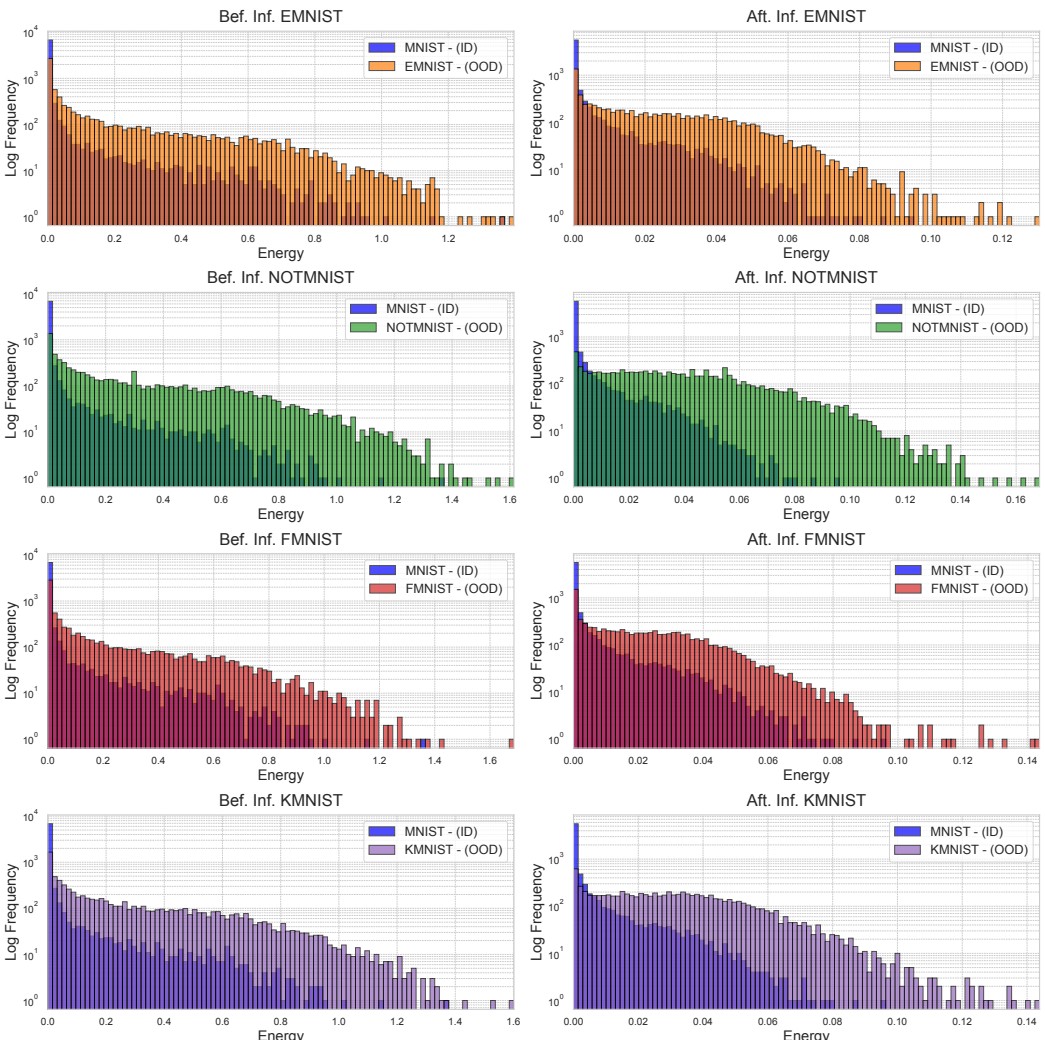

Figure 22: Energy histograms against ID data before and after state optimization.

In Fig. 22 we then show how each energy distribution for the OOD dataset compares against the energy of the in-distribution dataset by overlaying the histograms of the energies before and after state optimization. We can see that by plotting the histograms, a pattern emerges, namely, that a majority of the OOD data samples do not overlap with ID data samples, which supports the idea that energy can be used for OOD detection.

Next in Fig. 23 we show how this pattern might look like when comparing the softmax scores of ID against OOD datasets. One can see, that the softmax scores are less informative for determining if samples are OOD as can be seen by the bigger overlap in the range of softmax values that ID and OOD samples have in common.

In Fig. 24 we further study the relationship between softmax scores and energy values before and after state convergence. The plot shows that while the energy and softmax scores are strongly correlated before inference, a non-linear relationship is evident after convergence, especially for smaller values where the model is more uncertain. This indicates, that softmax scores and energy values do not fully agree on which samples we should have less confidence in.

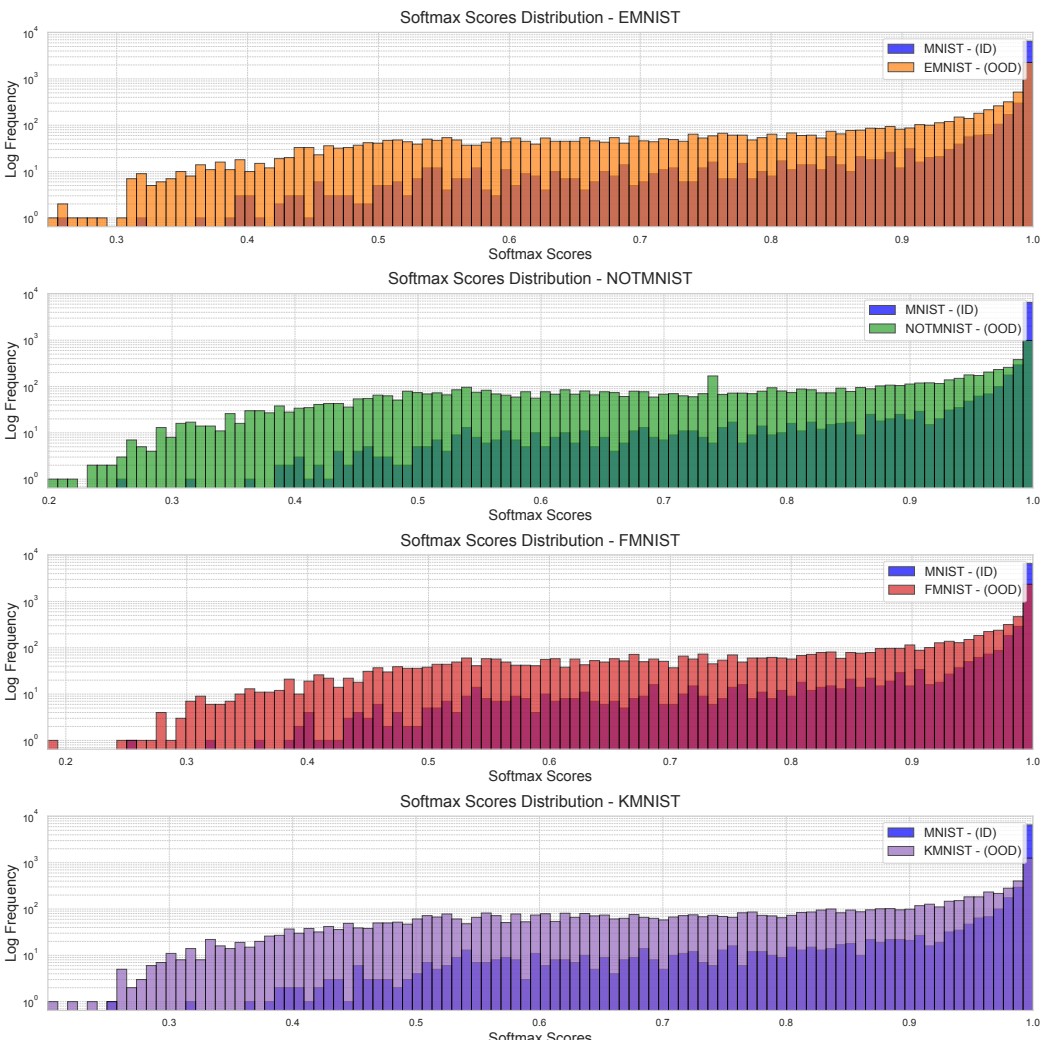

Figure 23: Softmax histograms overlapped with ID dataset.

In Fig. 25 we show how the energy distributions for all datasets look like before and after inference. Each box plot represents a different scenario and a different dataset. In addition, we compute the NLL of each dataset and display it as part of the box plot labels. We observe that across all OOD datasets, the initial and final energy values are significantly higher than the MNIST (ID) dataset. Furthermore, we can see that the variance of the energy scores is smaller for the in-distribution data as can be seen by the fact, that there are no outlier samples for MNIST beyond the whiskers of the box plot. Finally, the NLL values for each scenario confirm this observation, with the likelihood of the MNIST data being significantly higher than that of the OOD distributions.

Finally, in Fig. 26 we show how the PCN can be used to classify samples as belonging to the ID or some OOD data. We use the PCN classifier's energy to perform OOD detection and we show that the ROC curves for energy-based detection are superior to ROC curves created via softmax scores. This observation becomes even clearer, when looking at the most challenging samples by picking the 25% percentile of the scores and energies, in effect the samples, that the PCN model is least confident about as reflected by small energy or softmax values.

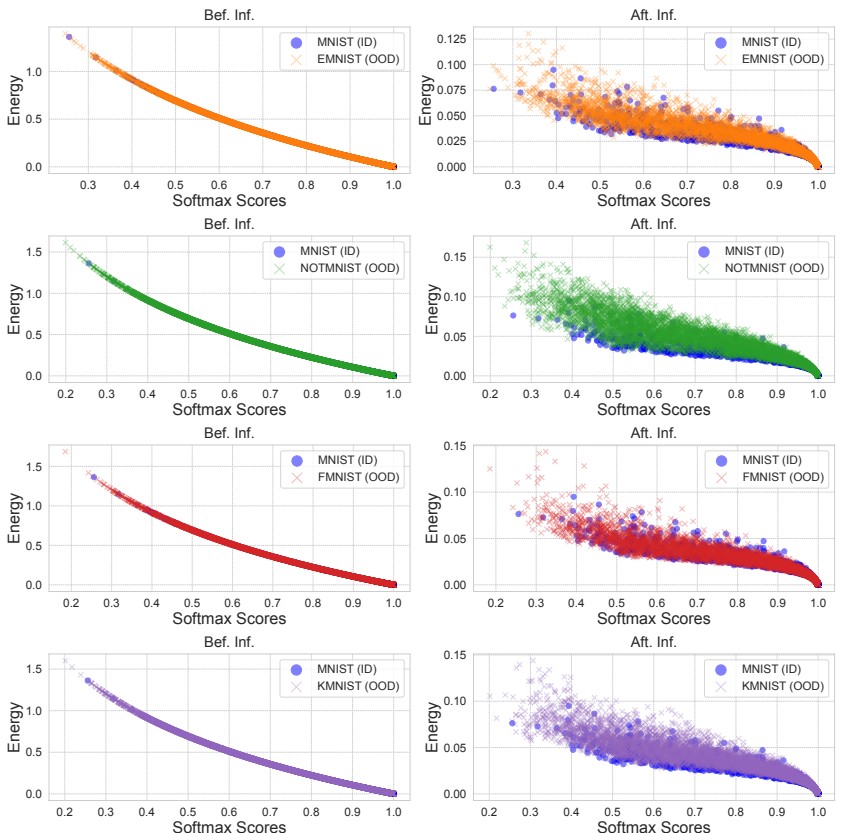

Figure 24: Non-linear relationship between energy and softmax scores.

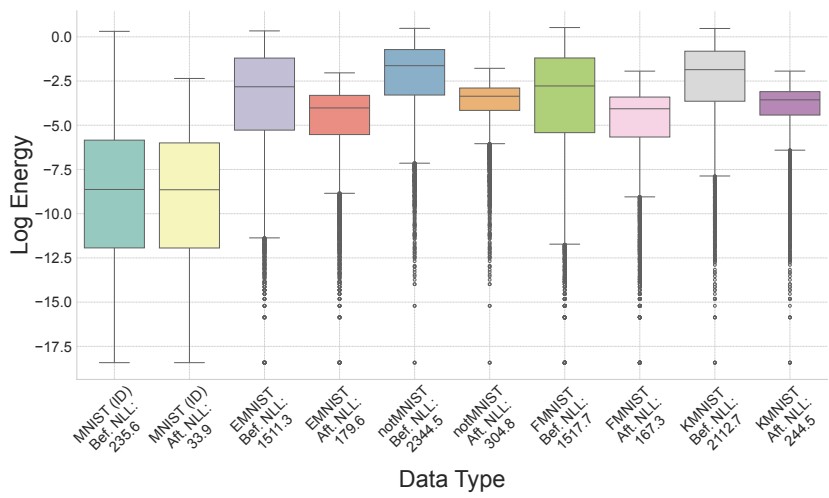

Figure 25: Energy and NLL for various OOD datasets before and after inference.

# G  COMPUTATIONAL RESOURCES

Fig. 8 was obtained by taking a small feedforward PCN made by 2 layers of 64 neurons each and training it on batches of 32 elements (generated as random noise so to avoid any overhead due to loading training data to the GPU) for $T = 8$ steps. Then, each parameter was scaled independently to measure its effect on the total training time. Each model obtained this way was trained for 5

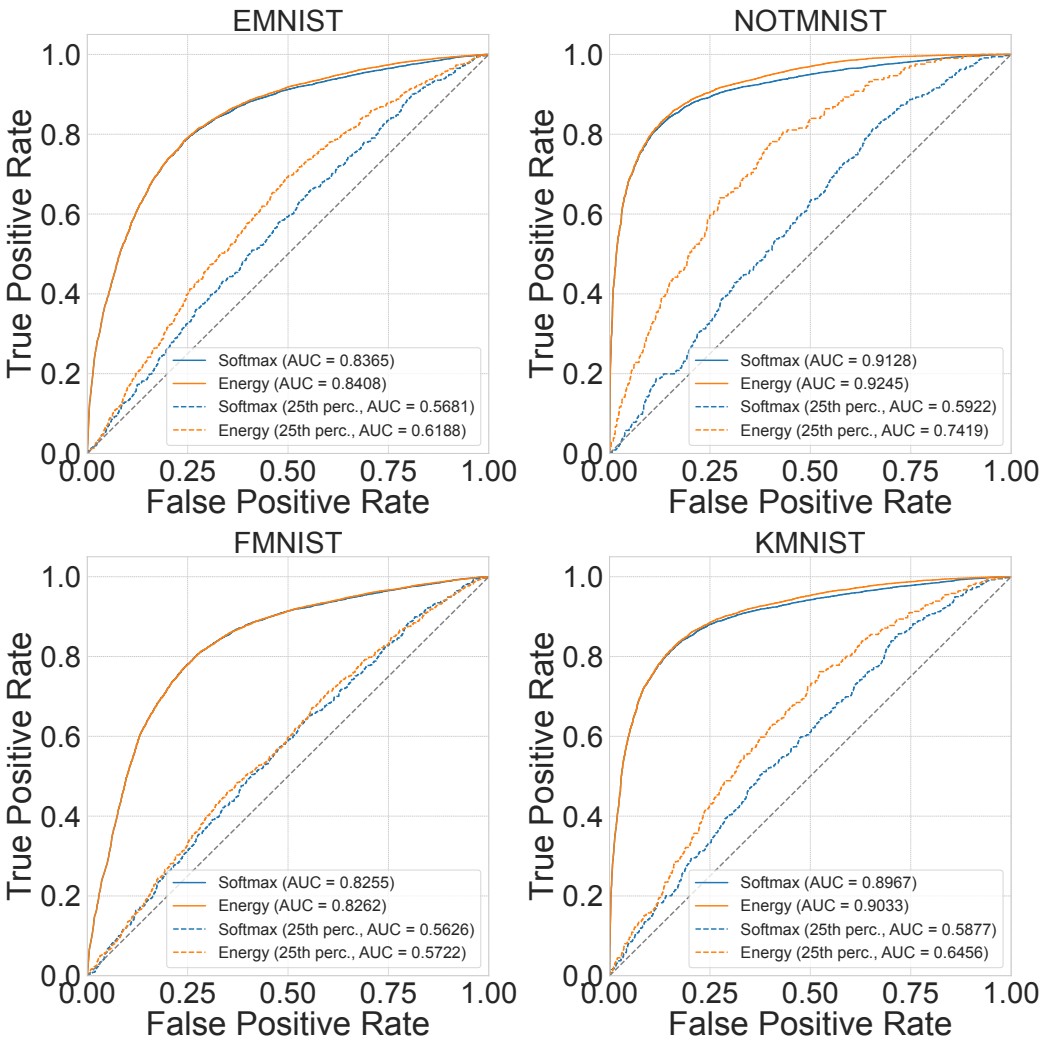

Figure 26: Performing OOD detection with PCN energy and classifier softmax scores.

epochs and the mean time was reported. In all our timing measurements, we skip the first epoch to avoid including the JIT compilation time. Results were obtained on a GTX TITAN X, showing that parallelization is potentially achievable also on consumer GPUs.

