# OpenReview forum: "Benchmarking Predictive Coding Networks -- Made Simple"
_ICLR.cc/2025/Conference — ICLR 2025 Spotlight_

### Official Review · Reviewer_9Atp · 2024-10-29

**Soundness:** 2
**Presentation:** 2
**Contribution:** 3
**Rating:** 6
**Confidence:** 3

**Summary:**

This paper addresses the trend in predictive coding networks (PCNs) literature of not comparing results against each other. To this end, the authors present a library that facilitates implementing and testing new PCNs, and they also propose several new benchmarks for evaluation. These benchmarks include larger networks and more complex tasks than those in previous works, and span discriminative as well as generative tasks. The paper evaluates several existing methods, including standard backpropagation, on these newly proposed benchmarks and analyzes the results.

**Strengths:**

### Originality
The main contribution of the paper is the PCX library which can be used to implement predictive coding networks, and the proposed benchmarks, which can be used to evaluate the scalability of new predictive coding networks.

### Quality
The PCX library, owing to Just-In-Time compilation, is much faster than the baseline library. The reported performance (training times) of the PCX library shines against the baseline by Song et al.

### Clartity
The tutorials are of good quality. The Jupyter Notebook tutorials released with the paper are easy to set up and follow.

### Significance
The paper is well motivated.
1. Providing the community with the tools and benchmarks to compare their methods against others’ is a meaningful endeavour.
2. The benchmarks are designed to address scalability of PCNs.
3. The benchmarks are not limited to only discriminative tasks like classification, but also include generative tasks such as autoencoding and sampling from a learned joint distribution.

The following insights into PCNs were very interesting:
1. The AdamW optimizer performs poorly as the hidden layer size increases. This was also demonstrated well in the tutorial notebook.
2. Skip connections have a dramatic effect on the performance of PCNs.

The evaluation is extensive. A fair number of existing PCN methods were evaluated (standard PC, positive/negative/centered nudging PC, incremental PC, Monte Carlo PC).

**Weaknesses:**

I gave a score of 3 for Contribution but only 1 for Soundness. This is because I believe the PCX library is a valuable contribution to the community; however, I have reservations about some of the conclusions and claims made in the paper.

Consider the Energy propagation subsection of Section 5.1 (Energy and Stability). L377 says, “Note that the decay in performance as function of increasing γ is stronger for Adam despite being the overall better optimizer in our experiments. This suggests that possible directions for future improvements should aim at reducing the energy imbalance between layers.”. It is unclear to me how this conclusion that energy imbalance must be reduced was arrived at. I understand energy imbalance to be different from energy ratios in that the imbalance is minimized when the ratio is 1, i.e., both the layers have the same energy.
In Figure 5 (right), we see that the learning rate which minimizes the energy imbalances (i.e, make the ratio as close to 1 as possible) is 0.3. In Figure 5 (center right), we see that the performance of AdamW optimizer for a learning rate of 0.3 is much worse than 1e-3, 3e-3, and 1e-2, all of which have much higher energy imbalances. The same pattern can be observed in Figures 13 and 14. How do we conclude from this behaviour of AdamW that the energy imbalance must be minimzed? This also affects the claim in the abstract that the paper “clearly highlights what the current limitations of PCNs are”.

Next, in Section 4.1 (Discriminative Mode), L243 says that "The results show that the best performing algorithms, at least on the most complex tasks, are the ones where the target is nudged towards the real label, that are PN and NN.". The results for Discriminative mode are reported in Table 1. Assuming, based on L067, that complex tasks refer to CIFAR100 and Tiny ImageNet, I am unable to find data in the table that substantiates this claim. For VGG-5 models, centered nudging (CN) dominates both positive nudging (PN) and negative nudging (NN) for all the tasks except Tiny ImageNet (Top-1).
For VGG-7 models, all PC methods are beaten by either of the two variants of Back Propagation (BP-CE and BP-SE). Even considering only the PC methods, PN is consistently beaten by PC-CE, as is NN, with the exception of CIFAR-100 (Top-1).

Finally, in the Discussion subsection of Section 4.2 (Generative Mode), line 339 claims that “Thus, compared to artificial neural networks, PCNs are more flexible and require only half the parameters to achieve similar performance”. The fact that PCNs require only half as many parameters as an autoencoder neural network is evident because they can infer the hidden states of layers that have not been fixed (either to the input image for compression or to the compressed representation for generation). However, the paper demonstrates that this holds true only for small models, as the generative experiments were conducted exclusively on relatively small models (3 layers for autoencoding and generation via sampling, and 2 hidden layers for associative memory experiments). Conducting at least one experiment on a larger model, such as a VGG-7-scale model, could have strengthened this claim.

I am open to reconsidering my scores if these concerns are addressed.

The following is feedback and minor nitpicks about the presentation of the paper. Addressing these will improve the quality of the paper, but will not affect the scores.

1. The abbreviation “BP” is used in the paper without introducing it. The first reference to it is in the Appendix.
2. Figure 1 (b). Shouldn’t it be “hL is fixed to y”?
3. Table 3 can be made more intuitive to read by mentioning in the caption that it is a comparison of the two modes of associative memory tasks - Noise and Mask.
4. H and theta should be subscripts in L141
5. Citations are missing for several works (which, admittedly, a lot of us take for granted). Examples are CIFAR-10, MNIST, Fashion-MNIST, Tiny ImageNet, SGD, Adam, AdamW, VGG models.
6. Citations for each of the methods in the Algorithms subsection of Section 4 would make it easier for the readers to refer to the papers, even though they are all cited before.
7. Figure 5 would be better if it were indexed as Fig 5a to Fig 5d, instead of Fig 5 left, center-left, center-right and right.
8. The citation style in L239 is wrong.
9. L347 “How regularly the energy flows into the model”. Improve phrasing.
10. All references to Figure 5 right and center-right are inverted. See, e.g., L373 and L377.
11. Grammatical errors in a few places, such as L267, L355, L467

**Questions:**

Can you motivate the use of VGG-5 and VGG-7 models in the benchmarks? In terms of model sizes, they are roughly 3.8M and 4.5M, respectively, so they’re quite close to each other.

---

> ### Author Response · Authors · 2024-11-21
>
> We thank the reviewer for the time and feedback, and also for the large list of typos/ small comments on the manuscript. They have all been addressed (please see the updated manuscript for these). We have also used the three major concerns raised to further improve the quality of the manuscript, mostly caused, we believe, from a lack of clarity/bad phrasing of sentences from our side We discuss them below:
>
>
>
>
> > Concern in Section 4.1: CN performs better than other nudging methods.
>
> This is easy to address, as it is caused by a typo from our side, for which we apologize. More in detail, the algorithms that use nudging are PN, NN, and CN. In the original sentence, we omitted CN, and only mentioned PN and NN as nudging algorithms. We have addressed this, and also added a more comprehensive explanation (partially guided by your concern). The new sentence now reads as follows:
>
> “Table 1 shows that the best results for PC models, at least on the most complex tasks, are achieved using nudging algorithms (PN, NN, and CN). Among them, CN is almost always the best performing one, a result that is in line with previous findings in the Eqprop literature [CITATION] . The only case where nudging algorithms are outperformed is on Tiny Imagenet on VGG7, where PC-CE performs better than them. However, the results obtained by PC-CE here, are still worse than the ones obtained by CN on VGG5”
>
> This sentence is now entirely true: all of the best results on the three image datasets have been reached using one of the three nudging algorithms. In all but two cases, CN was the best performing one, in the other two, NN performed better. The only case where nudging algorithms are outperformed is on Tiny Imagenet on VGG7, where PC-CE performs better than them. However, the results obtained by PC-CE here, are still worse than the ones obtained by CN on VGG5.
>
>
>
>
> > Concern on Section 4.2 (The fact that PCNs require only half as many parameters as an autoencoder neural network is evident):
>
> We agree with the reviewer that the discussion about the number of parameters was misleading, and, in all honesty, not so influential in terms of memory requirements given the size of the models we have used. It has now been removed. This applies to the sentence you have highlighted, and to figure 2, that was confronting our methods against both autoencoders with the same number of parameters as the PCN, and autoencoders with half of the number of parameters. Also, mentions of it at the beginning of the section have also been removed.

---

> > ### Author Response · Authors · 2024-11-21
> >
> > > Concern on Section 5.1
> >
> > We believe that this misunderstanding is also from poor phrasings from our side. In particular, when we wrote “This suggests that possible directions for future improvements should aim at reducing the energy imbalance between layers.”, we were referring to one sentence before: “While models trained with large \gamma values achieve better energy propagation, they achieve lower accuracy as shown in Fig5 (right).”
> >
> > The crux of the issue is that an energy ratio significantly less than 1 between layers results in exponentially vanishing weight gradients. Our claim is that current training solutions for PCNs optimize for the best results on small-scale networks but poorly scale to deeper architectures (as we show in Section 4), because they suffer from a “vanishing energy” (and thus gradient) problem due to a low state learning rate $\gamma$. In other words, the energy remains accumulated in the last layers and does not propagate effectively to layers closer to the input, which consequently receive very little training signal. Thus, obtaining a more balanced energy propagation is fundamental to apply PC to deeper networks.
> >
> >  More in general, the assumption that the gradients’ magnitude must remain approximately constant throughout the network is a requirement to ensure training stability and avoid vanishing and exploding gradients [1].
> >
> > We have re-written the paragraph completely, that now reads as follows:
> >
> > “To better understand how the energy propagation relates to the performance of the model, we have analyzed both the test accuracy and the ratio of the energies of subsequent layers as a function of the state learning rates $\gamma$. The results, reported in Fig~\ref{fig:s5}(c,d), show that small learning rates lead to better performance, but also to large energy imbalances among layers. On the one hand, the energy in the first hidden layer is similar to that of the last layer for $\gamma=1$, and about 6 orders of magnitude lower for $\gamma=0.01$. On the other hand, models trained with a learning rate of $\gamma=1$ achieve much worse performance. Such results show that the current training setup favors large energy imbalances among different layers, a problem that leads to exponentially small gradients when the depth of the model increases.”
> >
> > With regards to the experiments with AdamW, both in the “Energy propagation” and “Training stability” sections, we wanted to further highlight the issue, showing how, despite achieving the best accuracy overall on the model tested, AdamW does so for a set of hyperparameters (e.g., state learning rate \gamma) that are theoretically suboptimal when scaling to larger sizes (and, furthermore, is simply unstable with larger layer sizes or when using higher values for the state learning rate. Conditions which are both necessary to train wider and deeper networks).
> >
> >
> >
> > [1] Glorot, Xavier, and Yoshua Bengio. "Understanding the difficulty of training deep feedforward neural networks." Proceedings of the thirteenth international conference on artificial intelligence and statistics. JMLR Workshop and Conference Proceedings, 2010.

---

> > > ### Comment · Reviewer_9Atp · 2024-11-25
> > >
> > > I thank the authors for their response.
> > >
> > > > This is easy to address, as it is caused by a typo from our side …
> > >
> > > The updated sentence is accurate.
> > >
> > > > … not so influential in terms of memory requirements given the size of the models we have used. This has now been removed …
> > >
> > > Thank you. The paper is better without this claim.
> > >
> > > > Section 5.1 … We have re-written the paragraph completely …
> > >
> > > The new paragraph is much more clear and theoretically justifies the flaw in the existing training setups. I understand that the two main findings in the paper concerning the flaws of existing training setups are that (1) lower learning rate harms energy propagation and (2) the AdamW optimizer displays poor performance for larger hidden dimensions. In the light of these findings, it would have been worthwhile to empirically demonstrate that training a larger model with these conditions removed (E.g., using medium to high learning rates, and SGD optimizer instead of Adam), performs better than the same model trained under the current setup (even if the results are not better than a model of roughly the same size trained with BP).
> > >
> > > I notice that the authors have addressed the other minor feedback, with the exception of two:
> > > 1. The citation style in (now) line 247 is still incorrect.
> > > 2. The authors use Adam and AdamW interchangeably, and cite Kingma et al. [1] for AdamW. Please note that Adam and AdamW are from two different papers. The former is indeed by Kingma et al. AdamW is from Loshchilov and Hutter [2].
> > >
> > > [1] Diederik P. Kingma and Jimmy Ba, 2014. Adam: A Method for Stochastic Optimization
> > >
> > > [2] Ilya Loshchilov and Frank Hutter, 2017. Decoupled Weight Decay Regularization.
> > >
> > > Since my main concerns regarding the soundness of its claims have been addressed, I am happy to increase my score from 5 to 6.

---

> ### Author Response · Authors · 2024-11-28
>
> Dear Reviewer,
> Thank you for your comments. We have addressed the two typos remaining.
>
> About the ResNets experiments with SGD and large learning rates, we agree that they would be a nice addition to our work.  While we are running them at the moment, I feel we will not be able to report the complete study before the end of today (manuscript updates close in a couple of ours). However, we are happy to report results for PC and iPC, that have completed their large hyperparameter search. We have added them in the supplementary material, at the end of Page 26. More in detail, we have added a paragraph stating the following:
>
> **ResNets** *Here we discuss the findings on the energy propagation in light of the ResNets18 experiments. In this section, we have shown that lower learning rate for the nodes harm energy propagation, and that the AdamW optimizer displays poor performance for larger hidden dimensions. To this end, we have trained ResNets18 using SGD and large learning rates for the nodes, and compared the performance against those in the main body of the paper. The performance are, however, not comparable to the ones reported in Table.1, as ResNets trained with SGD on the CIFAR10 dataset reach accuracies of $39.9\%$ and $43.2\%$ when using PC and iPC, respectively. To better understand the incidence of different hyperparameters on the final test accuracy of the models, in Fig.~\ref{fig:importance} we show their importance plots. Such quantities are computed by fitting a random forest regressor with hyperparameters as datapoints, accuracies as labels, and extracting the feature importance.*
>
> As the text states, we have also computed, for every experiment with both SGD and adamW, the importance plots of every hyperparameter (the image mentioned in the paragraph, that is Figure 18 in the supplementary material).
>
> Note that this is just a placeholder paragraph and figure: we will have a full run of all models and algorithms in the following days, and will use them to make a much more comprehensive section/paragraph with the complete study. However, if we can judge the initial results, it seems that AdamW still outperforms SGD.
>
> Thank you again for the suggestion, we hope that this last change might improve your view of our paper’s soundness, and to consider it above a borderline score.
>
> Best,
>
> The authors

---

### Official Review · Reviewer_oRBc · 2024-11-01

**Soundness:** 3
**Presentation:** 3
**Contribution:** 3
**Rating:** 8
**Confidence:** 4

**Summary:**

The paper contributes a comprehensive library/framework that unifies the developing and testing for predictive coding networks (PCNs). Traditionally, the field suffers from lacking of unified metrics and open-source implementations, the contributed library well addresses the problem. Furthermore, with the contributed library, the paper is able to scale existing works onto larger datasets and achieves new SOTA on all the datasets. The paper also includes the popular algorithms in the field and benchmarked them extensively.

**Strengths:**

1. The benchmark results included in this work is very extensive which includes most of the popular baselines.
2. The design and usage of the contributed library are well documented with provided repos (I didn't run the code in this repo).
3. The analysis provided in the paper is well explained such as showing different results from different methods which makes the paper easy to read
4. With the contributed library, the paper is able to scale up the field to larger datasets which greatly improved the applicability of the methods in the field.
5. The work also shared some insights for sub-optimal cases of PCNs which can further guide the development of new methods.

**Weaknesses:**

1. Some figures can be improved, e.g. for figure 2, the method name can be draw above the images for better representation.
2. The main models used in the paper is MLP and VGG, since the library has been developed, we should probably test a few more model types such as resnet (shallow is OK if deep ones cannot be test) or vit. This could probably real more insights on the overall states of PCN algorithms.

**Questions:**

1. Can we test some models beyond VGG and MLP as mentioned in weakness?
2. For the results in table 1, the result of VGG-5 is better VGG-7, does that mean PC works better when the number of layers is small. I think we need more data points here.
3. Also for table 1, can you include the same datasets for both VGG-5 and VGG-7, currently, the missing data points (CIFAR-10) makes it both hard to read the table and hard to make more in-depth comparison between these two cases.
4. I think it's better to provide the URLs at the beginning of the paper instead of at the end.

I will consider raising the scores if my questions can be addressed.

---

> ### Author Response · Authors · 2024-11-21
>
> We thank the reviewer for his/her time and valuable feedback. In the updated manuscript, we have addressed all of the concerns. We also discuss them below.
>
>
> > Add experiments on Resnets:
>
> Thank you for this feedback, it has been addressed: as you can see in the updated manuscript, Table 1 is now much larger. We have added also results of our experiments on ResNets18. As the Table shows, the results are disappointing, as they are not comparable to those of smaller models, such as VGG7. We believe this to be the main area of improvement we should tackle. To do so, we require two directions of research: the first one should aim to understand why PCNs perform as well as BP on VGG7 models, and much worse on ResNets18; the second one should use this knowledge to then improve the models.
>
> > For the results in table 1, the result of VGG-5 is better VGG-7, does that mean PC works better when the number of layers is small. I think we need more data points here.
>
> Thank you for the suggestion: we have added more datapoints as requested. In detail, we have trained VGG9s on all of the datasets considered, and studied the decrease in performance. As shown in the updated manuscript (Table 1), VGG-9 models consistently underperform compared to VGG-5 models on the same tasks and are almost always outperformed by VGG-7 models. In summary, we observe the following performance hierarchy for PC-trained models: VGG-5 > VGG-7 > VGG-9. Conversely, for models trained with BP, the hierarchy is reversed: VGG-9 > VGG-7 > VGG-5. We have added a brief discussion of these findings to the manuscript, as well as summarizing the findings in a new figure (Fig.2), that shows how for CIFAR10 the above argument holds (including ResNets). Note that the argument holds for all of the datasets, and all of the training algorithms; we have added a figure for CIFAR10 only due to a lack of space.
>
>
> >  (1)  Include CIFAR10 on VGG-7; (2) Provide the URLs at the beginning of the paper; (3) Improve Figure 2.
>
>
> Done, thank you for the pointers!

---

### Official Review · Reviewer_sAME · 2024-11-03

**Soundness:** 4
**Presentation:** 3
**Contribution:** 4
**Rating:** 8
**Confidence:** 4

**Summary:**

The authors undertake the very hard task of trying to streamline, standardise and robustify the scientific process in the subfield of Predictive Coding networks. They identify that the subfield suffers from a variety of issues such as poor comparisons between models, too simple baselines or benchmarks, and overall a lack of rigour in how potential proposed techniques are studied, compared, contrasted and analysed. This leads to the issue of making important issues hard to identify, and even harder to get a good critical mass of people working on them for resolution.

The authors try to improve on this issues by offering:

1.  A unified benchmark for predictive coding network models that is more expansive than others before it, utilising CIFAR100, Tiny ImageNet, FashionMNIST and MNIST.
2. A library that allows quick and easy application of the benchmark on a target approach.
3. A thorough evaluation of a number of key architectures and PC approaches on their benchmark, that can serve both a source of insights for the current state of the field and a good starting baseline for future approaches.

Overall the proposal of the authors is an important and valuable one, and while the writing is OK, it is at times sloppy, with fluency and spelling mistakes, as well as inconsistent use of various terms.

However overall a very good paper.

**Strengths:**

- Addresses Core Issues in PC Research: The paper tackles fundamental shortcomings in the subfield, such as inconsistent comparisons, overly simple benchmarks, and a lack of rigorous, standardized practices.

- Unified Benchmarking Framework: Introduces a comprehensive and standardized benchmark for PCN models, incorporating major datasets like CIFAR-100, Tiny ImageNet, FashionMNIST, and MNIST to ensure consistent and meaningful comparisons.

- Implementation of Open-Source Tool (PCX): Offers a JAX-based library that enhances research efficiency and accessibility, boasting up to 50x speed improvements via JIT compilation and seamless compatibility with modern ML tools.

- Thorough Experimental Evaluation: Conducts extensive tests on key PCN architectures and variants, backed by large-scale hyperparameter studies, establishing a solid baseline for future advancements.

- Deep Analysis of Training Mechanics: Provides valuable insights into energy propagation, initialization strategies, and stability across optimizers and architectures, which can guide targeted future research.

- Community-Centric Contribution: The open-source nature, complete with detailed documentation and tutorials, lowers the barrier to entry and encourages collaboration, setting a strong foundation for collective progress.

- Motivation for Future Research: Clearly identifies current limitations, such as scalability and energy flow issues, laying the groundwork for targeted solutions and deeper exploration.

- Competitive Performance on Benchmarks: Shows that PCNs can achieve performance on par with BP-trained models for small to medium-sized architectures, establishing a base for scaling up and further innovation.

Insightful 'weaknesses' highlighted by the paper that should not be perceived as a weakness of the paper itself imho.

Identification of Scalability Challenges: The paper highlights scalability issues inherent to PCNs but does not propose new solutions. While this reflects well on the depth of the analysis, it may be seen as a limitation in terms of contribution.
Performance Bottlenecks in PCNs: The analysis shows that while the proposed library offers performance improvements over prior implementations, it remains slower than backpropagation for more complex models. This insight underscores the current challenges facing PCNs but could be perceived as a weakness in terms of efficiency.
Sequential Processing Constraints: The authors identify that sequential layer updates are a significant bottleneck, impacting parallelism and efficiency. This is a critical finding for the field, though it might be seen as a shortcoming in practical use.
Theoretical Insight Depth: The paper provides an analysis of training mechanics, but it could go further in offering deep theoretical explanations for issues like energy propagation and training stability. This gap invites further exploration but could be seen as a weakness by readers looking for comprehensive theoretical coverage.
Implementation Constraints and Parallelization: The paper highlights that training time scales linearly with the number of layers due to current parallelization limitations. This insight is crucial for understanding the limitations of PCNs but might be perceived as a practical constraint in the work presented.

**Weaknesses:**

Weaknesses of the Paper:
- Writing and Terminology: The paper has inconsistencies in terminology and occasional grammar and spelling issues, along with unclear phrasing that affects readability and comprehension.
- Comparative Analysis Limitations: The paper could be strengthened by including more comparisons with other biologically inspired approaches and better situating its findings within the broader machine learning context.
- Practical Implications: The paper lacks a thorough exploration of how its findings translate to practical, real-world applications beyond academic benchmarks.
- Diversity of domains/tasks/modalities while a lot more than anything other out there for PC research, is still very limited compared to much more general benchmarks out there for general deep learning evaluation. Adding more domains, tasks and modalities would be very useful to having rigorous and insightful works done by people using it.
- Architectures applied do not include any resnet, densenet or transformer architectures -- yes depth is an issue I understand, but using a shallow variant of any of these would be very useful.

**Questions:**

1. Why only JAX? Why not Pytorch? It seems like this goes against the spirit of the inclusiveness of your work.
2. On Scalability Solutions: While the paper does an excellent job of identifying scalability issues in PCNs, do the authors have any preliminary ideas or suggestions for future work that could address these challenges?

3. Efficiency vs. Backpropagation: Given that the proposed PCX library is still slower than traditional backpropagation for complex models, are there any plans or ongoing work to bridge this efficiency gap?
3. Spelling, grammar and clarity points:

Here are the sections that could benefit from improved clarity, fluency, or grammar:

1. Line 110-111:
"different directions of research that either explored interesting properties of PC networks, such as their robustness and flexibility (Song et al., 2024; Alonso et al., 2022) , or proposed variations to"
- Extra space before comma after parenthesis
- Somewhat awkward phrasing

2. Lines 180-182:
"In practice, we do not train on a single pair (x,y) but on a dataset split in mini-batches that are subsequently used to train the model parameters."
- Should be "split into mini-batches"

3. Line 229:
"While interesting and important for the progress of the field,"
- Awkward start to sentence, could be rephrased for better flow

4. Line 270:
"We test the performance of PCNs on image generation tasks."
- Somewhat abrupt transition, could use better connection to previous section

5. Line 394:
"We observed a further link between the weight optimizer and the structure of a PCN that might hinder the scalability of PC"
- Run-on sentence that could be clearer if split or restructured

6. Line 443:
"PCX extensively relies on Just-In-Time compilation."
- Inconsistent capitalization of "Just-In-Time" compared to other instances of "just-in-time"

7. Line 469:
"In fact, in its current state, JIT is unable to parallelize the executions of the layers;"
- "executions" should be singular "execution"

8. Line 476:
"we have lied new foundations"
- Should be "laid new foundations"

9. Line 485:
"When this condition is released,"
- Should be "When this condition is relaxed," or "When this limitation is removed,"

10. Throughout the paper:
- Inconsistent spacing around mathematical symbols and equations
- Some inconsistent use of commas in complex sentences
- Occasional mixing of American and British English spellings

Figure 1:

You want a figure to stand on its own without any context from the paper ideally. What is F here? Can we improve the clarity and informativeness of the caption and the figure itself?

These issues are relatively minor and don't significantly impact the paper's readability or scientific contribution, but addressing them would improve the overall polish of the manuscript.

---

> ### Author Response · Authors · 2024-11-21
>
> We thank the reviewer for his/her time and valuable feedback.
>
> > Writing, terminology, spelling, and clarity points:
>
> Thank you for the extensive list of pointers. We have addressed them in the final version of the manuscript. We have also had another full pass on the manuscript that focused on clarity.
>
> > The paper could be strengthened by including more comparisons with other biologically inspired approaches.
>
> Initially, we had considered doing so, especially as the accuracies we reach are often comparable, or better, than the ones reached by other neuroscience-inspired methods, and hence would look good for us. However, in the end we have decided to avoid the comparison, as different methods have different, interesting, properties, and hence some of them may be preferred in some cases, despite reaching worse test accuracies. For example, equilibrium propagation methods may sometimes  be preferred to PC due to existing implementations on some kind of neuromorphic hardware (https://www.nature.com/articles/s41928-022-00869-w); this is despite them reaching slightly worse results on more complex datasets. In other words, we would like to avoid a “ranking” of bio-plausible methods, and only focus on the one this paper is about. We have however carefully cited the related works of other biologically plausible methods, highlighting which ones present the best results: the reader interested in such a comparison can refer to them.
>
> > Diversity of domains/tasks/modalities
>
> We agree that this work does not come close to the modern machine learning world in terms of ‘diversity’ of domains, tasks, and modalities, and that the more we add, the more valuable our work would be. We are actively working on it, but with respect to this specific paper, we are leaving it for future work. We also hope that such a gap could be filled with the help of a community effort: what we have provided in this work (library, tutorials, etc.) lowers the entry barrier for researchers in the field to take initiatives and implement their own benchmarks/tasks.
>
> > Why JAX and not Pytorch?
>
> This library has been internally developed, refined, and tested for a significant amount of time. We opted for JAX for two main reasons:
>
> 1) Initially, PyTorch support for compilation was not very mature, and JAX seemed a more reliable choice, also in terms of existing documentation and possible future support.
>
> 2) As we also have shown in this work, there are many versions of predictive coding, each with its own pros and cons, and we believe that more and more variations will be necessary to scale this technology up. JAX is considered by many as having a more “low-level” approach and, as such, a lot of the libraries developed for it, such as equinox, diffrax, or optax, allow for more refined control that may not be necessary for backpropagation-based neural networks, but may ease the research and development of more ad-hoc techniques (such as predictive coding).
>
> Also, according to our personal experience and to online comments, it seems that as of today, JAX’s jitting is still the faster of the two (but it seems a bit task dependent). However, it could be an interesting future research direction to compare the two, as PyTorch uses a different compiler, which could reveal itself to be more suited for predictive coding.
>
>
> > On Scalability Solutions, as well as the addition of experiments on ResNets.
>
> Thank you for the suggestion. In the manuscript, we have added a section that shows the results of our experiments on ResNets18. As the Table shows, the results are disappointing, as they are not even close to those of smaller models, such as VGG5 or 7. We believe this to be the main area of improvement we should tackle. To do so, we require two directions of research: the first one, that aims to understand why PCNs perform as well as BP on VGG7 models, and much worse on ResNets18. Then, we have to use this understanding by either adapting the algorithm used, or modifying the architecture (One example could be the addition of  more skip connections). We have started to address the first direction of research in section 5.1, where we try to shed some light on a kind of “vanishing gradient” problem that seems to be present in deeper models; as well as proposing different avenues of improvements in the supplementary material, such as the addition of backward connections that help in the spreading of the error.
>
>
> > Given that the proposed PCX library is still slower than traditional backpropagation for complex models, are there any plans or ongoing work to bridge this efficiency gap?
>
>
> Yes, we are working on parallelizing the operations that would allow us to parallelize the backward messages (the errors). This would allow deeper models to train deep models faster, at the cost of a larger memory complexity. This tradeoff will (hopefully) be customizable by the user.

---

> ### Author Response · Authors · 2024-11-28
>
> Dear Reviewer,
>
> We would like to flag another addition to the paper:
>
> We are running a large hyperparameter search to study the performance of ResNets trained with SGD, as our study of the energy propagation suggests that this may lead to some interesting findings ( as suggested by Reviewer 9Atp).  We are happy to report the results for PC and iPC, that have completed their large hyperparameter search, and will add the remaining ones in the following days. We have added them in the supplementary material, at the end of Page 26. More in detail, we have added a paragraph stating the following:
>
> **ResNets** *Here we discuss the findings on the energy propagation in light of the ResNets18 experiments. In this section, we have shown that lower learning rate for the nodes harm energy propagation, and that the AdamW optimizer displays poor performance for larger hidden dimensions. To this end, we have trained ResNets18 using SGD and large learning rates for the nodes, and compared the performance against those in the main body of the paper. The performance are, however, not comparable to the ones reported in Table.1, as ResNets trained with SGD on the CIFAR10 dataset reach accuracies of $39.9\%$ and $43.2\%$ when using PC and iPC, respectively. To better understand the incidence of different hyperparameters on the final test accuracy of the models, in Fig.~\ref{fig:importance} we show their importance plots. Such quantities are computed by fitting a random forest regressor with hyperparameters as datapoints, accuracies as labels, and extracting the feature importance.*
>
> As the text states, we have also computed, for every experiment with both SGD and adamW, the importance plots of every hyperparameter (the image mentioned in the paragraph, that is Figure 18 in the supplementary material).
>
> Note that this is just a placeholder paragraph and figure: we will have a full run of all models and algorithms in the following days, and will use them to make a much more comprehensive section/paragraph with the complete study. However, if we can judge the initial results, it seems that AdamW still outperforms SGD.
>
> Thank you again for your time,
>
> Best,
>
> The authors

---

### Author Response · Authors · 2024-11-21

Dear Reviewers,

Thank you for taking the time and effort to review our work. While we have provided individual responses to each of your comments, we would like to use this opportunity to discuss some of the key revisions made to the manuscript, many of which were inspired by your insightful feedback:

**VGG-9:**
To further investigate how the performance of PCNs changes with increased network depth, we conducted an extensive hyperparameter search using VGG-9 models. This additional analysis provides further evidence supporting our claim that performance differences between PCNs and models trained with BP become more pronounced as depth increases. As shown in the updated manuscript (Table 1), VGG-9 models consistently underperform compared to VGG-5 models on the same tasks and are almost always outperformed by VGG-7 models. In summary, we observe the following performance hierarchy for PC-trained models: VGG-5 > VGG-7 > VGG-9. Conversely, for models trained with BP, the hierarchy is reversed: VGG-9 > VGG-7 > VGG-5. We have added a brief discussion of these findings to the manuscript, as well as summarizing the findings in a new figure (Fig.2), that shows how for CIFAR10 the above argument holds (including ResNets). Note that the argument holds for all of the datasets, and all of the training algorithms; we have added a figure for CIFAR10 only due to a lack of space.


**ResNets:**
We also extended our analysis by training multiple ResNet-18 models on the same tasks. The results strongly reinforce the trend observed with VGG models: test accuracies achieved using PCNs are significantly lower, with none of the ResNet models approaching the performance of a VGG-5. In contrast, ResNet-18 models trained with BP outperform all previously tested VGG models, further highlighting the gap in performance between the two training paradigms.

Best regards,
The authors

---

### Meta-Review · Area_Chair_gCS1 · 2024-12-18

**Metareview:**

The paper makes several contributions to predictive coding networks in machine learning in the form of a library to enable and facilitate comparisons, an attempt at unification of benchmarks, as well as rigorous empirical evaluation. These latter points address several identified pain points and open aspects in existing literature. As such, all reviewers agree that the paper is valuable, necessary, and will likely have impact in the community. The AC agrees with these points and believes the paper presents a timely contribution to advance the field. The raised weaknesses were mostly addressed in the discussion period. Overall, the AC recommends to accept the paper and agrees with the reviewers that the paper’s contributions are extensive and the library helpful to the community.

**Additional Comments On Reviewer Discussion:**

Raised points for improvement included several clarifications in form of writing and presentation, a request for extended empirical corroboration, alongside several raised questions on more open aspects and future work. During the discussion phase, the authors have made several additions to the paper and have managed to further improved the manuscript. This was well received by the reviewers.

---

### Decision · Program_Chairs · 2025-01-22

Accept (Spotlight)